# H3K9me2 orchestrates inheritance of spatial positioning of peripheral heterochromatin through mitosis

Andrey Poleshko[1], Cheryl L Smith[1], Son C Nguyen[2], Priya Sivaramakrishnan[2], Karen G Wong[1], John Isaac Murray[2], Melike Lakadamyali[3], Eric F Joyce[2], Rajan Jain[1,4,5]*, Jonathan A Epstein[1,4,5]*

[1]Department of Cell and Developmental Biology, Perelman School of Medicine, University of Pennsylvania, Philadelphia, United States; [2]Department of Genetics, Perelman School of Medicine, University of Pennsylvania, Philadelphia, United States; [3]Department of Physiology, Perelman School of Medicine, University of Pennsylvania, Philadelphia, United States; [4]Department of Medicine, Perelman School of Medicine, University of Pennsylvania, Philadelphia, United States; [5]Penn Cardiovascular Institute and Institute of Regenerative Medicine, Perelman School of Medicine, University of Pennsylvania, Philadelphia, United States

*For correspondence:
jainr@pennmedicine.upenn.edu (RJ);
epsteinj@pennmedicine.upenn.edu (JAE)

Competing interests: The authors declare that no competing interests exist.

**Abstract** Cell-type-specific 3D organization of the genome is unrecognizable during mitosis. It remains unclear how essential positional information is transmitted through cell division such that a daughter cell recapitulates the spatial genome organization of the parent. Lamina-associated domains (LADs) are regions of repressive heterochromatin positioned at the nuclear periphery that vary by cell type and contribute to cell-specific gene expression and identity. Here we show that histone 3 lysine 9 dimethylation (H3K9me2) is an evolutionarily conserved, specific mark of nuclear peripheral heterochromatin and that it is retained through mitosis. During mitosis, phosphorylation of histone 3 serine 10 temporarily shields the H3K9me2 mark allowing for dissociation of chromatin from the nuclear lamina. Using high-resolution 3D immuno-oligoFISH, we demonstrate that H3K9me2-enriched genomic regions, which are positioned at the nuclear lamina in interphase cells prior to mitosis, re-associate with the forming nuclear lamina before mitotic exit. The H3K9me2 modification of peripheral heterochromatin ensures that positional information is safeguarded through cell division such that individual LADs are re-established at the nuclear periphery in daughter nuclei. Thus, H3K9me2 acts as a 3D architectural mitotic guidepost. Our data establish a mechanism for epigenetic memory and inheritance of spatial organization of the genome.
DOI: https://doi.org/10.7554/eLife.49278.001

## Introduction

In order for a dividing cell of a given lineage to maintain its identity, it must pass along to its progeny not only a complete copy of its genome, but also the memory of its specific cellular identity (*Buchwalter et al., 2019*; *Towbin et al., 2013*; *Amendola and van Steensel, 2014*). It is well appreciated that the spatial arrangement of the genome inside the nucleus contributes to regulation of cell-fate choices and differentiation (*Peric-Hupkes et al., 2010*; *Phillips-Cremins et al., 2013*). However, the mechanistic underpinnings of how the blueprint for cell-type-specific nuclear architecture is transmitted from mother to daughter cells in order to maintain cell identity remain poorly understood (*Dekker et al., 2017*).

The chromatin in eukaryotic cells is organized both structurally and functionally into subnuclear compartments (*Towbin et al., 2013*; *Kohwi et al., 2013*; *Stadhouders et al., 2019*) and recent

developments in super-resolution microscopy (*Cremer et al., 2017*; *Ricci et al., 2017*), chromosome capture methods (*Dekker et al., 2002*; *Dekker et al., 2013*), and chromatin immunoprecipitation (ChIP) (*Collas, 2010*; *Kubben et al., 2010*) have greatly increased our understanding of 3D nuclear architecture (*Naumova et al., 2013*). Separation of transcriptionally active and inactive chromatin in three-dimensional space reinforces efficient regulation of gene expression and maintains silencing of heterochromatic loci (reviewed in *Andrey and Mundlos, 2017*; *Buchwalter et al., 2019*; *Amendola and van Steensel, 2014*; *Bickmore, 2013*). This is illustrated by examples of aberrant gene expression patterns that occur upon disruption of topological domains and, in extreme cases, are associated with oncogenic transformation (*Andrey and Mundlos, 2017*; *Flavahan et al., 2016*). Heterochromatin is segregated into spatially distinct subnuclear compartments including peripherally located lamina-associated domains (LADs) (*Guelen et al., 2008*), which encompass approximately 30–40% of the genome (*Peric-Hupkes et al., 2010*; *Poleshko et al., 2017*). Multiple examples in mammalian cell types indicate that proper positioning of LADs contributes to cell-type-specific gene expression (*Peric-Hupkes et al., 2010*; *Poleshko et al., 2017*; *Robson et al., 2016*). Likewise, in Drosophila, competence of neuroblasts to respond to inductive signals depends upon stage-specific reorganization of peripheral heterochromatin (*Kohwi et al., 2013*), and muscle differentiation in *Caenorhabditis elegans* requires anchoring of heterochromatin to the nuclear periphery (*Gonzalez-Sandoval et al., 2015*). These findings, combined with the observation that many developmental and lineage-specific genes reside in LADs, suggest a key role for peripheral heterochromatin in establishment and maintenance of cellular identity (*Zullo et al., 2012*; *Poleshko et al., 2017*; *Peric-Hupkes et al., 2010*). LADs are defined by their interaction with the nuclear lamina which is disassembled during cell division, posing a conundrum as to how cell-type specific LADs are remembered through mitosis.

The molecular mechanisms by which LADs are established and maintained at the nuclear periphery remain poorly understood. For example, there does not appear to be a clear targeting sequence that localizes areas of the genome to the nuclear periphery (*Zullo et al., 2012*; *Meuleman et al., 2013*). However, histone post-translational modifications have been implicated in LAD regulation. Proline Rich Protein 14 (PRR14) has been shown to recognize H3K9me3, found on both peripheral and nucleoplasmic heterochromatin, through an interaction with HP1 (*Poleshko et al., 2013*). In addition, work from our group and others has demonstrated a specific enrichment for H3K9me2 at the nuclear periphery, raising the possibility of a regulatory role in LAD positioning (*Poleshko et al., 2017*; *Kind et al., 2013*). CEC-4, a *C. elegans* chromodomain-containing protein, localizes to the nuclear periphery and has been shown to be a reader of H3K9 methylated chromatin (*Gonzalez-Sandoval et al., 2015*). Depletion studies using RNAi and loss-of-function mutants demonstrated that CEC-4 is required for peripheral heterochromatin anchoring but not transcriptional repression. While not all of the tethering complexes and molecular determinants responsible for the interaction of heterochromatin with the nuclear lamina have been determined, it is clear that these associations must be disrupted upon mitotic entry when the nuclear envelope breaks down and the chromosomes condense. Furthermore, these interactions must be precisely re-established upon mitotic exit when the cell reforms an interphase nucleus.

Entry into mitosis involves eviction of proteins, including RNA polymerase and many transcription factors, and reorganization of chromosomes into their characteristic metaphase form (*Naumova et al., 2013*). Remarkably, at mitotic exit, cell-type-specific chromatin architecture, transcription factor binding, and gene expression are re-established (reviewed in *Oomen and Dekker, 2017*; *Palozola et al., 2019*; *Hsiung and Blobel, 2016*; *Probst et al., 2009*; *Festuccia et al., 2017*). While both interphase nuclear architecture and post-mitotic restoration of transcription factor association with the genome have been extensively studied (*Palozola et al., 2019*; *Kadauke and Blobel, 2013*), our understanding of how cell-type-specific genome organization including LADs is restored in daughter cells after mitosis is less well developed.

Pioneering studies in the 1980 s revealed the necessity for DNA in the process of nuclear lamina reassembly after mitosis, and the activity of kinases and phosphatases were implicated in mediating interactions between lamin and chromosomes (*Foisner and Gerace, 1993*; *Newport, 1987*; *Burke and Gerace, 1986*; *Gerace and Blobel, 1980*), although the mechanistic explanation for the dependence of reassembly on chromatin has been unclear. Here, we utilize high resolution, single-cell imaging and oligopaints to simultaneously track 82 LAD and non-LAD genomic loci through mitosis. We show that the H3K9me2 modification of nuclear lamina-associated heterochromatin,

revealed upon dephosphorylation of H3S10 at mitotic exit, provides a 3D spatial guidepost for genomic regions that are to be re-localized to the nuclear periphery following mitosis and that the nuclear lamina of daughter cells reassembles around the exposed H3K9me2 mark.

## Results

### H3K9me2 is an evolutionarily conserved mark of peripheral heterochromatin

Heterochromatin is organized in multiple compartments throughout the nucleus (*Pueschel et al., 2016*), and H3K9me2 is a posttranslational histone modification that specifically marks heterochromatin at the nuclear periphery (*Poleshko et al., 2017*). Immunostaining of murine NIH/3T3 fibroblasts for repressive histone modifications demonstrates the distribution of the major types of heterochromatin in the nucleus of a single cell (*Figure 1a*). H3K9me2 marks only peripheral heterochromatin, whereas H3K9me3 and H3K27me3 co-localize with heterochromatin in the nuclear interior, or at both the interior and the periphery (*Figure 1a*, *Figure 1—figure supplement 1*). The close association between H3K9me2 and the nuclear lamina marker Lamin B in single cell immunostaining is consistent with the correlation between H3K9me2 and Lamin B ChIP-seq data (*Figure 1—figure supplement 1*). The adjacency of H3K9me2 chromatin to the nuclear lamina was verified by super-resolution microscopy (*Figure 1b*). Stochastic Optical Reconstruction Microscopy (STORM) using a Voronoi tessellation confirms a non-random distribution of the H3K9me2 signal at the periphery of the nucleus (*Figure 1—figure supplement 2*). We further examined H3K9me2-marked heterochromatin across species and observe that restriction to the nuclear periphery is evolutionarily conserved from *C. elegans* to humans (*Figure 1c*) suggesting functional significance of the localization of this histone post-translational modification.

Previously, distinctions between genomic regions marked by H3K9me2 versus H3K9me3 were unclear, perhaps because of lack of specificity of relevant antibodies. Therefore, we extensively characterized the specificity of the H3K9me2 antibody employed in these studies (*Figure 2*, *Figure 2—figure supplement 1*). By preincubating the anti-H3K9me2 antibody with peptides representing each of the possible histone tail modifications before use in immunostaining, we were able to determine that the H3K9me2 antibody detects only the dimethyl modification and only on lysine 9 of histone H3 (*Figure 2a*, *Figure 2—figure supplement 1*). Additionally, by blocking the H3K9me2 antibody with an H3K9me2 peptide, the specific signal observed at the nuclear periphery can be distinguished from non-specific background signal observed in the nuclear interior and detected with signal intensity analysis (*Figure 2b*). This observation was further confirmed by STORM imaging (*Figure 2c*).

### H3K9me2 is required for nuclear peripheral localization of chromatin

Given the specificity of H3K9me2 for peripheral heterochromatin, we hypothesized that this epigenetic histone modification is necessary for peripheral localization of chromatin and might be recognized by a nuclear peripheral protein 'reader' to tether chromatin to the nuclear lamina (*Figure 3a*). In *C. elegans,* CEC-4 functions as a reader of methylated H3K9 and is localized to the nuclear periphery where it is thought to function as part of a tethering complex for peripheral heterochromatin (*Gonzalez-Sandoval et al., 2015*). Mammalian functional orthologues of CEC-4 have not yet been identified. Since CEC-4 is required for peripheral heterochromatin anchoring (*Gonzalez-Sandoval et al., 2015*), we compared the localization of H3K9me2 in wild-type and *cec-4*-null embryo cells. Immunostaining revealed a dramatic alteration in spatial patterning in which H3K9me2 is no longer restricted to the periphery in *cec-4*-null cells (*Figure 3b and c*, *Figure 3—source data 1*). Localization of the H3K9me2-marked chromatin at the nuclear lamina was restored by expression of the CEC-4-mCherry transgene (*Figure 3c*, *Figure 3—figure supplement 1*). Despite previous observations of CEC-4 binding to all methylated forms of H3K9 in vitro (*Gonzalez-Sandoval et al., 2015*), in vivo loss of CEC-4 does not affect H3K9me3 localization. H3K9me3 is found both at the nuclear periphery and in the nucleoplasm, but its localization does not vary between wide-type and *cec-4*-null embryo cells (*Figure 3—figure supplement 1*). These data suggest loss of a peripheral heterochromatin tether, CEC-4, results in a specific effect on H3K9me2-marked chromatin and not H3K9me3-marked chromatin.

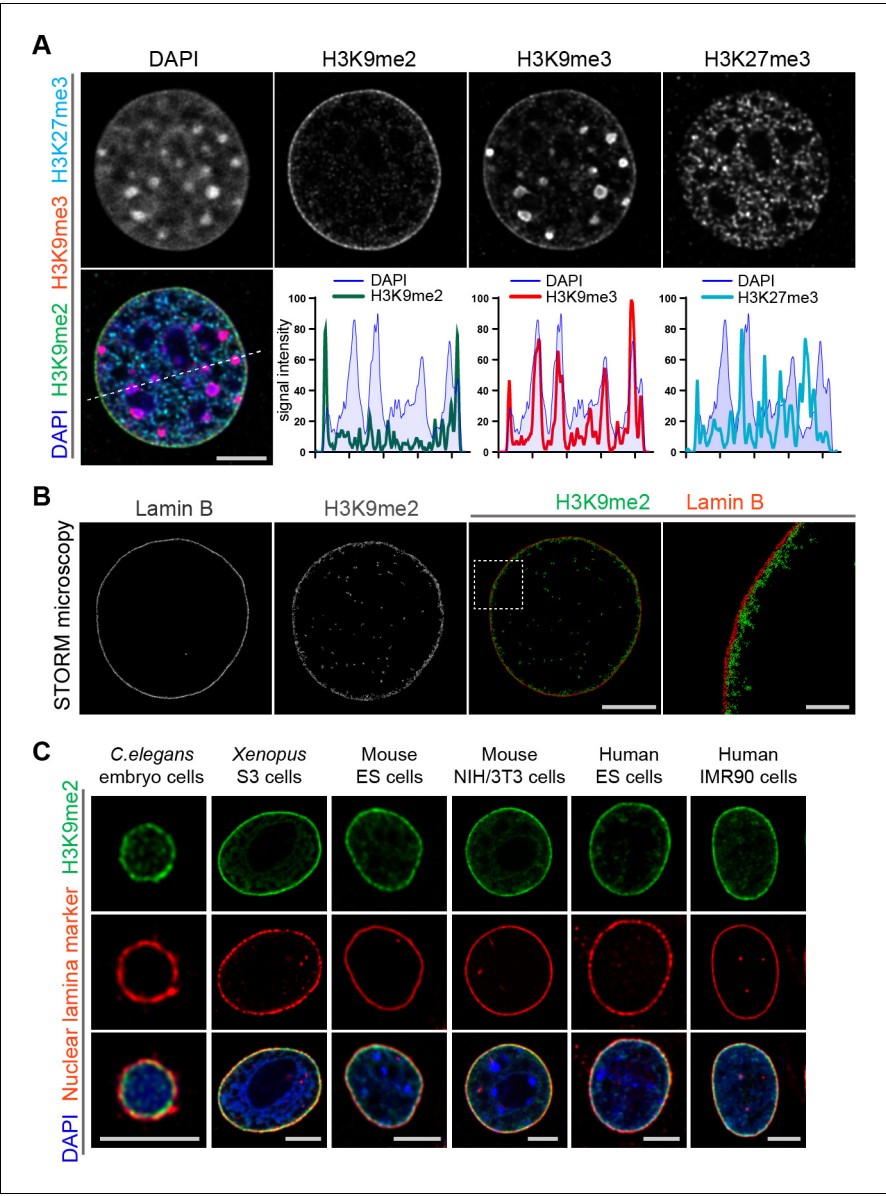

**Figure 1.** Localization of H3K9me2-marked chromatin at the nuclear periphery is evolutionarily conserved. (**A**) Immunofluorescent confocal images illustrating localization of the indicated repressive chromatin marks in the nucleus of a NIH/3T3 cell, counterstained with DAPI; dashed line indicates position of the line signal intensity profiles. Scale bar: 5 μm (**B**) Representative super-resolution images of a NIH/3T3 cell stained for H3K9me2 and Lamin B obtained using Stochastic Optical Reconstruction Microscopy (STORM). Scale bars: 5 μm (left panel) and 1 μm (right panel) (**C**) Localization of H3K9me2-marked chromatin in distinct species, co-stained with nuclear lamina markers (Lamin one for *C. elegans*; Lamin B all others), counterstained with DAPI. Scale bars: 5 μm.
DOI: https://doi.org/10.7554/eLife.49278.002

The following figure supplements are available for figure 1:

**Figure supplement 1.** H3K9me2-marked chromatin localizes specifically at the nuclear periphery and forms large heterochromatin domains.
DOI: https://doi.org/10.7554/eLife.49278.003

**Figure supplement 2.** H3K9me2 signal distribution is specific at the nuclear periphery.
DOI: https://doi.org/10.7554/eLife.49278.004

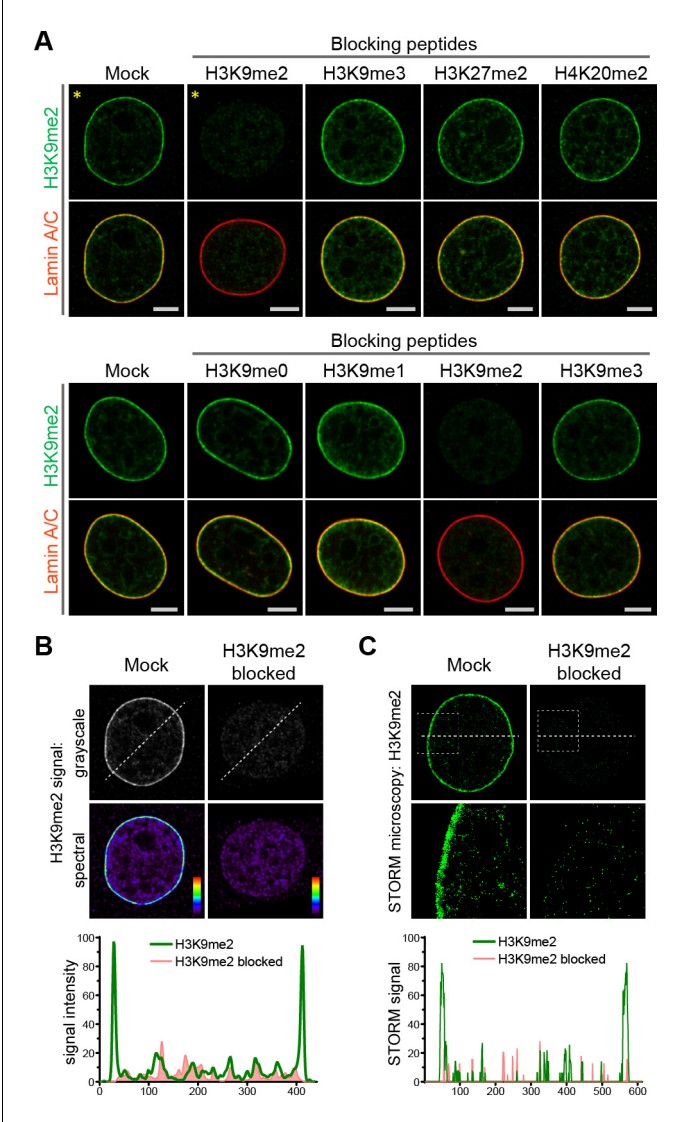

**Figure 2.** Anti-H3K9me2 antibody used in immunofluorescence assays is specific. (**A**) Murine C2C12 cells stained with nuclear lamina marker Lamin A/C and H3K9me2 antibodies preincubated with indicated blocking peptides. (**B**) Starred images (*) from panel A, with H3K9me2 signal displayed in grayscale and signal intensity spectral view; line signal intensity profile, below, illustrates H3K9me2-specific signal (green) and non-specific antibody background (red). (**C**) STORM images of NIH/3T3 cell stained for H3K9me2 and blocked with mock or H3K9me2 peptide; line signal intensity profile below as in panel B.

DOI: https://doi.org/10.7554/eLife.49278.005

The following figure supplement is available for figure 2:

**Figure supplement 1.** Anti-H3K9me2 antibodies validation.

DOI: https://doi.org/10.7554/eLife.49278.006

To extend our results and probe the role of H3K9 in chromatin positioning in mammalian cells, we expressed GFP-tagged histone H3 (hereafter H3) or GFP-tagged mutant forms of H3 in which Lys9 was substituted with alanine (H3K9A) or glutamic acid (H3K9E); both substitutions preclude methylation at this position in H3. GFP-tagged proteins were expressed in NIH/3T3 cells at relatively low levels compared to endogenous H3 (*Figure 3—figure supplement 2*) and attempts to drive higher levels of expression resulted in cell death. Wild-type GFP-H3 was observed throughout the nucleus including at the nuclear periphery, where it overlapped with endogenous H3K9me2 staining, immediately adjacent to Lamin B (*Figure 3d*). In contrast, GFP-H3K9A and GFP-H3K9E failed to

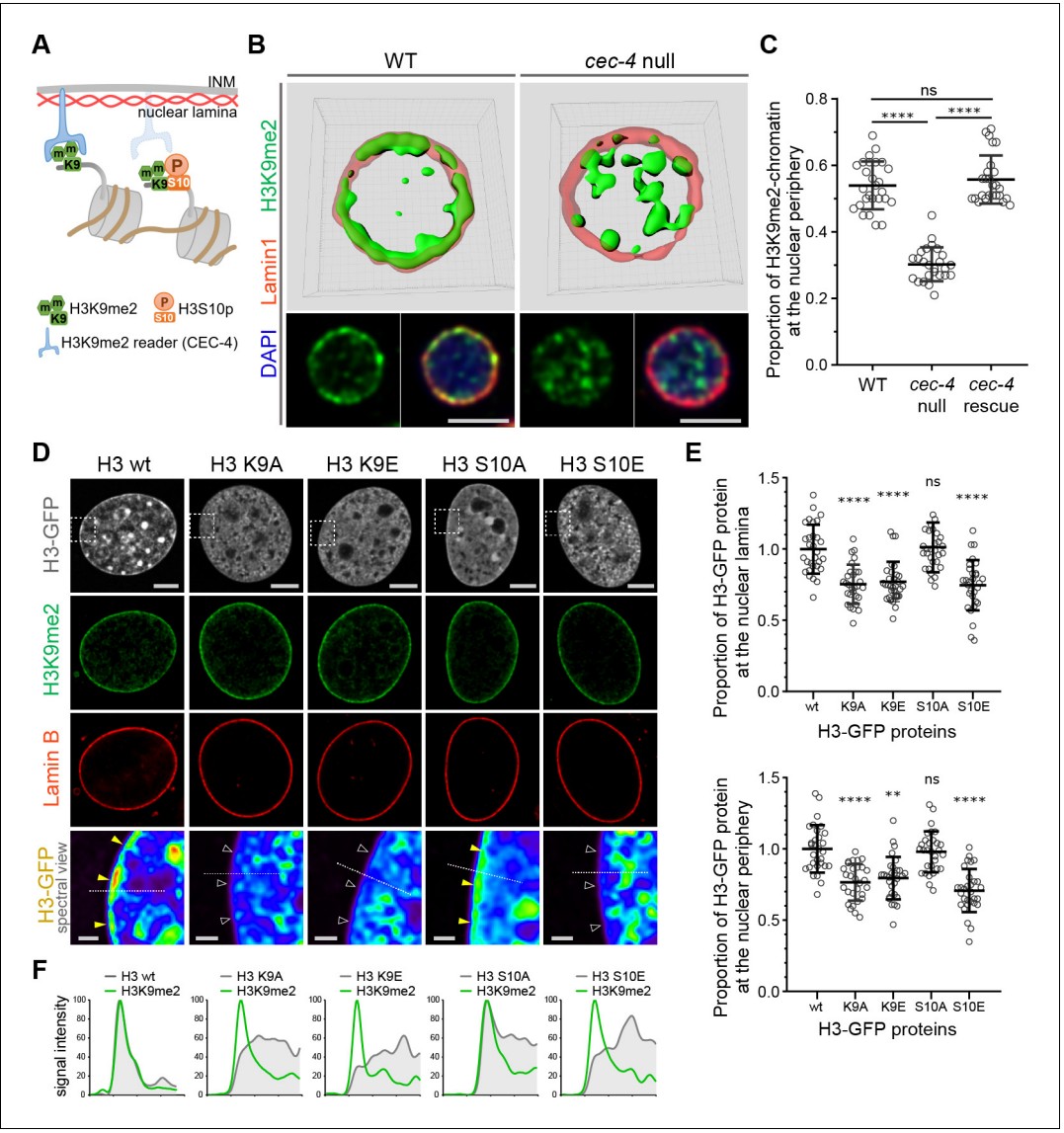

**Figure 3.** H3K9me2 is essential for histone H3 positioning at the nuclear periphery. (**A**) Schematic illustrating *C. elegans* protein CEC-4 tethering H3K9me2-marked chromatin to the nuclear periphery; INM: inner nuclear membrane. (**B**) Localization of H3K9me2-marked chromatin (green) in wild-type (WT) and *cec-4*-null *C. elegans* embryo cells, counterstained with nuclear lamina marker Lamin 1 (red) and DAPI (blue); 3D reconstruction (top); immunofluorescent confocal images of *C. elegans* embryo cells (bottom). Scale bars: 3 μm (**C**) Dot plot of the proportion of total H3K9me2-marked chromatin at the nuclear lamina in WT, *cec-4*-null, and *cec-4*-rescued embryo cells (mean ± SD); n = 25 cells per condition. (**D**) Localization of indicated histone H3-GFP fusion proteins in NIH/ 3T3 cells; counterstained with H3K9me2 (green) and nuclear lamina marker Lamin B (red); spectral views (magnifications of top panels as indicated by dashed squares) illustrate H3-GFP signal intensity. Localization of the H3-GFP at the nuclear periphery (yellow arrowheads) or loss of peripheral localization (white arrowheads). Scale bars: 5 μm (top panels) and 1 μm (bottom panels). (**E**) Dot plot of the proportion of indicated H3-GFP fusion protein at the nuclear lamina (marked by Lamin B, top) or within the layer of peripheral heterochromatin (marked by H3K9me2, bottom), normalized to wt H3-GFP, calculated using Lamin B or H3K9me2 signal as a mask (mean ± SD); n = 30 cells per condition. (**F**) Line signal intensity profiles of corresponding images in panel D indicated by dashed lines. Statistical analyses performed using two-tailed student's t-test for panel C and one-way ANOVA test for panel E; ****p<0.0001, **p=0.0024, ns: not significant; all comparisons relative to wild type (wt).
DOI: https://doi.org/10.7554/eLife.49278.007

The following source data and figure supplements are available for figure 3:

**Source data 1.** Numerical data related to *Figure 3C*.

*Figure 3 continued on next page*

*Figure 3 continued*

DOI: https://doi.org/10.7554/eLife.49278.010

**Source data 2.** Numerical data related to *Figure 3E*.

DOI: https://doi.org/10.7554/eLife.49278.011

**Figure supplement 1.** Localization of H3K9me2- and H3K9me3-marked chromatin in *C. elegans* wild-type (WT), *cec-4*-null, and *cec-4*-rescue embryo cells.

DOI: https://doi.org/10.7554/eLife.49278.008

**Figure supplement 2.** Expression of histone H3-GFP fusion proteins.

DOI: https://doi.org/10.7554/eLife.49278.009

partition to the nuclear periphery (*Figure 3d–f*, *Figure 3—source data 2*). Given that wild-type GFP-H3 is incorporated and observed at the nuclear periphery, we interpret the inability of the K9A and K9E mutants to partition to the periphery to suggest that lysine nine dimethylation is required for either incorporation into peripheral nucleosomes, or for retention within nucleosomes at the periphery. Combined with the CEC-4 results, this indicates that dimethylation of H3K9 orchestrates positioning of chromatin to the nuclear periphery.

## A phospho-methyl switch controls peripheral heterochromatin localization

H3S10 phosphorylation is associated with mitotic chromosome condensation (*Wei et al., 1999*; *Prigent and Dimitrov, 2003*) and, together with the neighboring Lys9 residue, has been proposed to function as a 'phospho-methyl switch' to modulate binding of H3 to effector proteins (*Varier et al., 2010*; *Fischle et al., 2003*; *Wang and Higgins, 2013*). Expression of a GFP-tagged H3 mutant in which Ser10 is replaced with the phospho-mimic glutamic acid (H3S10E) resulted in distribution of the GFP-H3S10E throughout the nucleus, but notably not at the nuclear periphery (*Figure 3d–f*). This is consistent with the ability of phosphorylated Ser10 to inhibit interaction of the reader with H3K9me2 and suggests that phosphorylation of Ser10 can prevent H3 peripheral localization. Replacement of H3 Ser10 with an alanine (H3S10A) precludes phosphorylation at this site and did not disrupt peripheral localization. Instead, H3S10A produced a pattern similar to wild-type GFP-H3 in interphase cells (*Figure 3d–f*). Together, these H3 mutant results suggest that H3K9me2 is required for localization of heterochromatin to the nuclear periphery. Further, they indicate that phosphorylation of Ser10 can prevent or disrupt this association as part of a phospho-methyl switch. Indeed, experimental results from the Gasser lab demonstrated that CEC-4 binds methylated H3K9 peptides and this binding is reduced by 2 orders of magnitude if the adjacent Ser10 is phosphorylated (*Gonzalez-Sandoval et al., 2015*).

## H3K9me2 persists through mitosis and associates with reassembling nuclear lamina in daughter cells at mitotic exit

Given the requirement for H3K9me2 to position heterochromatin at the nuclear lamina in interphase, we asked whether the H3K9me2 mark is maintained through cell division or if the histone modification is lost and re-acquired de novo in daughter cells. Examination of cells progressing through the consecutive phases of mitosis revealed persistence of H3K9me2 on mitotic chromatin (*Figure 4a*, *Figure 4—figure supplement 1*). Prior to disassembly of the nuclear lamina in prophase, H3K9me2-marked chromatin begins to detach from the nuclear periphery. Concordant with this detachment, we observe phosphorylation of Ser10 on the H3 tail adjacent to dimethylated Lys9 (H3K9me2S10p) beginning in prophase and persisting until late telophase (*Figure 4a and b*). Similar to the anti-H3K9me2 antibody (*Figure 2*, *Figure 2—figure supplement 1*), we carefully tested the specificity of the anti-H3K9me2S10p antibody used in these experiments and verified that it does not recognize the H3K9me2 epitope without an adjacent phosphate group on S10, nor does it recognize H3S10p alone (*Figure 4—figure supplement 2*). H3S10 phosphorylation in prophase may contribute to release of H3K9me2 readers/tethers (*Eberlin et al., 2008*; *Hirota et al., 2005*) and detachment from the nuclear periphery. Our data suggest that not every histone H3 Ser10 adjacent to H3K9me2 is phosphorylated since we observe some overlap of staining with the H3K9me2 and H3K9me2S10p antibodies.

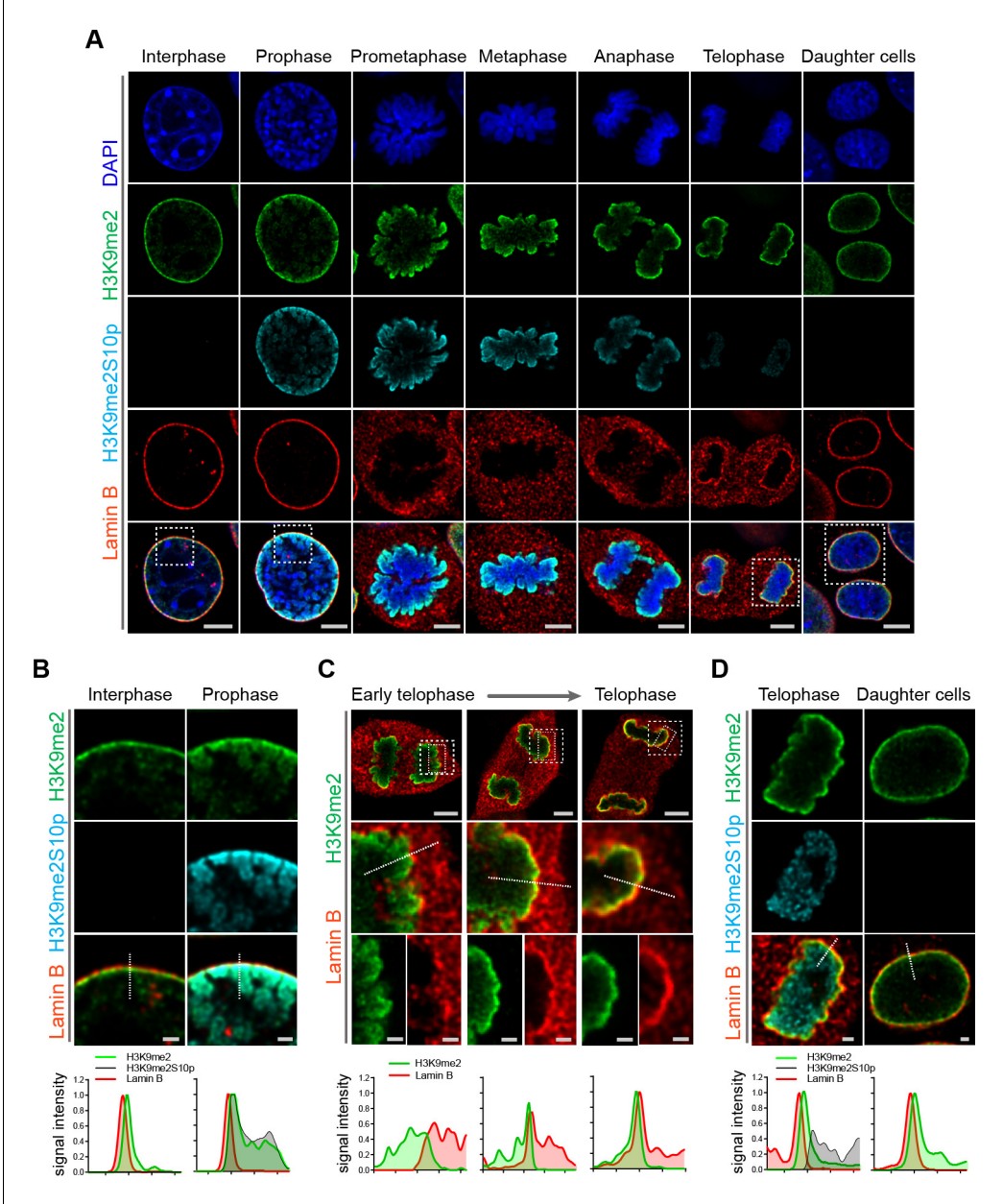

**Figure 4.** H3K9me2-marked chromatin is maintained throughout mitosis to be re-established at the nuclear lamina during nuclear lamina reassembly. (A) Representative immunofluorescent confocal images of murine C2C12 cells illustrating localization of H3K9me2- and H3K9me2S10p-marked chromatin and Lamin B during different stages of mitosis; DNA visualized with DAPI. Scale bars: 5 μm. (B) Magnified images of Interphase and Prophase from panel (A) demonstrating detachment of the H3K9me2-chromatin from the nuclear lamina concomitant with H3K9me2S10p phosphorylation; scale bar: 1 μm. (C) Representative images of cells progressing through telophase as the layer of peripheral H3K9me2-marked heterochromatin (green) is re-established and nuclear lamina (Lamin B, red) is reassembled; dashed boxes in top panels indicate higher resolution images. Scale bars: 5 μm (top) and 1 μm (bottom panels). (D) Magnified images of telophase and daughter cells from panel A demonstrating de-phosphorylated H3K9me2-chromatin (green) assembled at the nuclear lamina (Lamin B, red), while the phosphorylated form (H3K9me2S10p, cyan, enhanced brightness) remains localized in the nuclear interior; scale bar: 1 μm. Dashed lines indicate location of corresponding representative line signal intensity profiles (bottom row).

DOI: https://doi.org/10.7554/eLife.49278.012

The following figure supplements are available for figure 4:

*Figure 4 continued*

**Figure supplement 1.** 3D reconsruction of mitotic cells stained for H3K9me2.
DOI: https://doi.org/10.7554/eLife.49278.013
**Figure supplement 2.** Anti-H3K9me2S10p antibody specificity validation.
DOI: https://doi.org/10.7554/eLife.49278.014
**Figure supplement 3.** Restoration of the H3K9me2 chromatin layer at the nuclear lamina during telophase progression.
DOI: https://doi.org/10.7554/eLife.49278.015

We also examined cells at successive points in telophase. As telophase progresses, re-establishment of the H3K9me2 layer occurs in parallel with reassembly of the nuclear lamina. We observed aggregation of H3K9me2-marked chromatin and the reformation of this heterochromatin layer at the interface with the newly forming nuclear lamina structure (*Figure 4c*, *Figure 4—figure supplement 3*). However, chromatin marked with H3K9me2S10p was not enriched at the interface of the forming nuclear lamina but remained in the nucleoplasm (*Figure 4d*), suggesting that loss of S10 phosphorylation occurs prior to association of chromatin with the nuclear lamina. We detected little or no H3K9me2S10p in daughter cells after mitosis was complete (*Figure 4d*).

A subset of H3K9me3-marked chromatin is at the nuclear periphery, though it is not restricted to the periphery as is H3K9me2. H3K9me3 is enriched in microsatellite heterochromatin and persists through mitosis (*Figure 5a*). In addition, in telophase we noted strong differences in localization of other repressive (H3K9me3, H3K27me3) and active (H3K4me3) histone marks in contrast to H3K9me2 (*Figure 5b*). Trimethylated H3K9 is also distinct from H3K9me2 in that H3K9me3 chromatin is not enriched at the interface with the forming nuclear lamina during telophase and mitotic exit. In the newly formed daughter cells, we observed H3K9me2- but not H3K9me3-marked chromatin preferentially associated with the nuclear lamina.

## Specific LADs positioned at the nuclear periphery prior to mitosis re-associate with forming nuclear lamina in telophase

Restoration of H3K9me2-marked chromatin at the nuclear lamina prior to mitotic exit suggests a mechanism for inheritance of spatial localization of specific genomic loci within the peripheral heterochromatin layer. Our experiments thus far demonstrate that H3K9me2-marked chromatin, in general, is re-established at the nuclear lamina. Conflicting reports have emerged regarding whether LADs are stochastically reshuffled at every cell division or directed through a locus-specific, regulated mechanism to localize in other, non-lamina-associated heterochromatic subcompartments (*Kind et al., 2013*; *Zullo et al., 2012*; *Kind et al., 2015*). To determine whether specific genomic regions are re-established at the nuclear periphery at mitotic exit, we used fluorescence in situ hybridization (FISH)-based imaging to monitor the localization of individual genomic regions in single cells. We designed libraries of fluorescent DNA oligo probes (oligopaints) targeting domains of the genome that were identified through population-based studies (*Meuleman et al., 2013*; *Peric-Hupkes et al., 2010*; *Poleshko et al., 2017*) to be either cell-type invariant regions of nuclear peripheral, H3K9me2-marked heterochromatin (LADs) or cell-type invariant regions of euchromatin (non-LADs). The pool of probes (41 LAD and 41 non-LAD regions) includes regions from every mouse autosome (*Figure 6—figure supplement 1*, *Supplementary file 1*). We performed immunofluorescent in situ hybridization (immuno-FISH) with the probes in individual cells in interphase and mitosis; reconstruction of stacks of confocal images allowed us to visualize the 3D positions of each set of specific genomic loci (*Figure 6*, *Videos 1–3*).

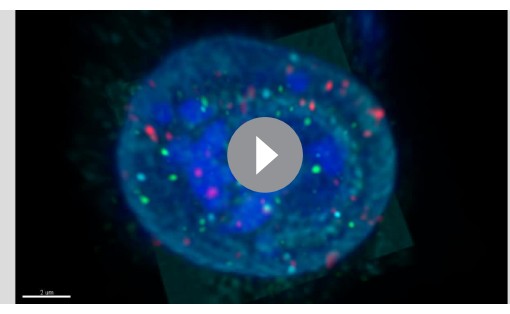

**Video 1.** 3D reconstruction of mESC in interphase. Immunostained for Lamin B1 (cyan) and hybridized with fluorescent oligopaint probes for LADs (red) and non-LADs (green), and counterstained with DAPI (blue).
DOI: https://doi.org/10.7554/eLife.49278.017

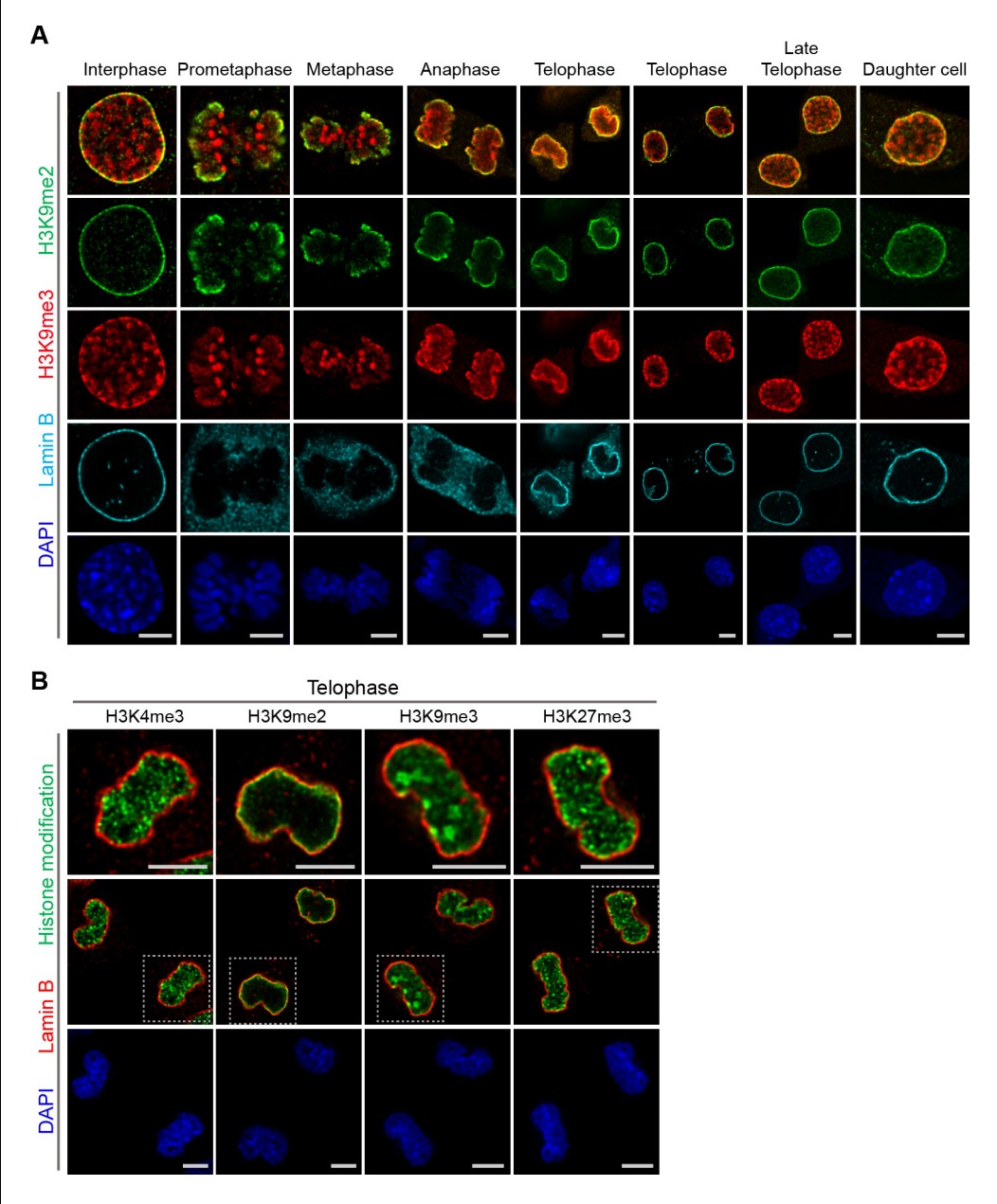

**Figure 5.** Localization of H3K9me2- and H3K9me3-marked chromatin differs during mitosis. (**A**) Representative immunofluorescent confocal images of murine C2C12 cells illustrating a difference in localization of H3K9me2 (green) and H3K9me3 (red) chromatin marks in interphase, during mitosis, and upon mitotic exit; co-stained with Lamin B (cyan) and DAPI (blue). (**B**) Representative immunofluorescent confocal images of C2C12 cells in telophase illustrating difference in localization of different histone modifications (green) in relation to Lamin B (red); co-stained with DAPI (blue). Dashed boxes in panels of middle row indicate higher resolution images (top row). Scale bars: 5 μm.

DOI: https://doi.org/10.7554/eLife.49278.016

In a population of interphase cells, we found the LAD probes to be at the periphery of individual nuclei at a frequency consistent with previous observations of haploid cells in studies using single-cell DamID (*Kind et al., 2015*). An average of 82% of LAD probes (74–90% in individual cells) were positioned at the nuclear periphery within the measured thickness of the H3K9me2 chromatin layer in interphase cells (*Figure 6a*, *Video 1*, *Figure 6—source data 1*). Non-LAD probes, assessed in

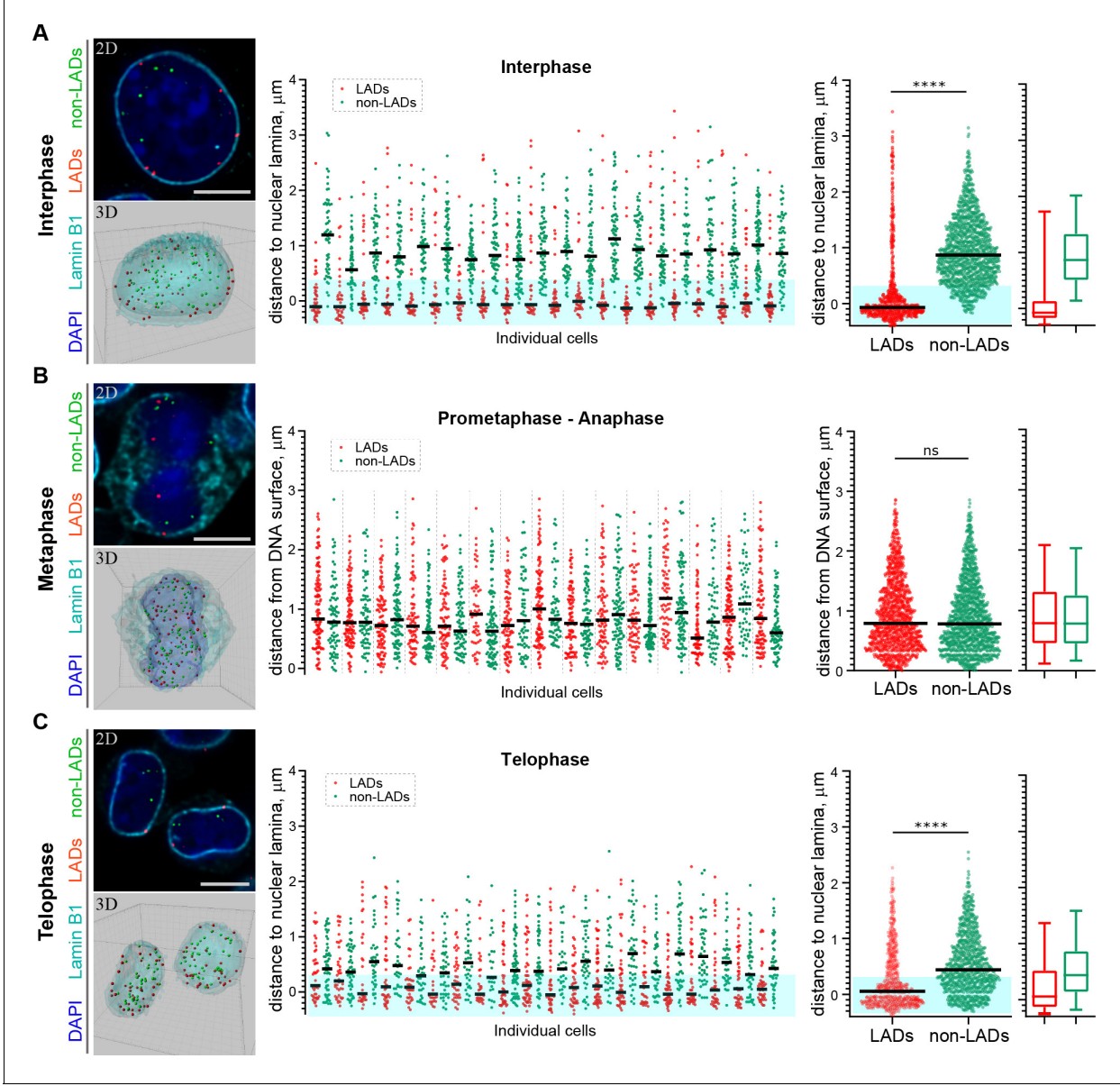

**Figure 6.** H3K9me2-enriched LADs are positioned at the nuclear lamina in interphase cells and the position is inherited through mitosis. (**A**) Localization of LADs and non-LADs in interphase mouse embryonic stem cells (mESCs). Left panels show representative immuno-FISH images (top) and 3D image reconstructions (bottom) of cells hybridized with fluorescent DNA oligopaint probes targeting individual LADs (red) and non-LADs (green), and immunostained for Lamin B1 (cyan) and DAPI (blue). Scale bar: 5 μm. Dot plots show distribution of distances to the nuclear periphery (as defined by Lamin B1) of individual LAD and non-LAD probes for individual cells (middle) and cumulative over all cells (right) in interphase. (**B**) As in panel A for prometaphase-metaphase-anaphase cells. (**C**) As in panel A for telophase cells. For dot plots, nuclear periphery defined by Lamin B1 or DNA edge; black line: median value; cyan boxes indicate average thickness of H3K9me2 peripheral heterochromatin layer. Box plots display 5, 25, 50, 75 and 95 percentiles. n ≥ 20 individual nuclei; N = 870–1399 individual LADs or non-LADs per condition. Statistical analysis performed using two-tailed t-test; ****p<0.0001; ns: not significant.

DOI: https://doi.org/10.7554/eLife.49278.020

The following source data and figure supplements are available for figure 6:

**Source data 1.** Numerical data related to *Figure 6*.
DOI: https://doi.org/10.7554/eLife.49278.023

**Figure supplement 1.** Location of the oligopaint DNA probes targeting LADs and non-LADs on mouse chromosomes.
DOI: https://doi.org/10.7554/eLife.49278.021

**Figure supplement 2.** Localization of LADs and non-LADs in interphase and mitotic mESCs.
DOI: https://doi.org/10.7554/eLife.49278.022

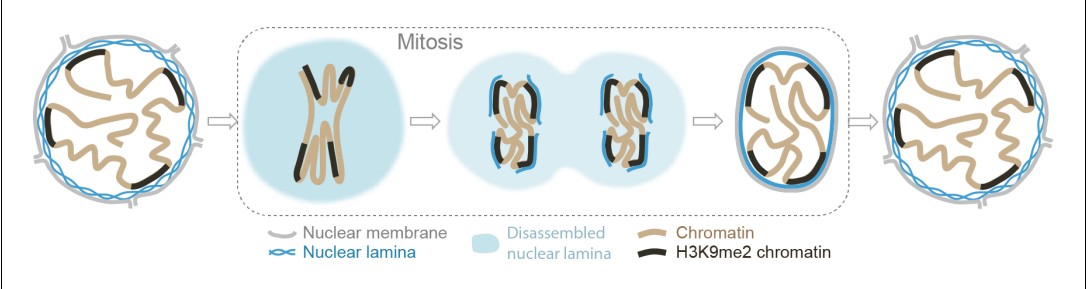

**Figure 7.** Model illustrating the role of the H3K9me2 chromatin modification in inheritance of peripheral heterochromatin localization through cell division.
DOI: https://doi.org/10.7554/eLife.49278.024

each of the same interphase cells, were more frequently found in the nucleoplasm, as expected: an average of 89% of non-LAD probes (79–95% in individual cells) segregated outside of the peripheral chromatin layer (*Figure 6a*, *Figure 6—source data 1*).

Next, we examined the location of these pools of representative LAD and non-LAD genomic loci in cells undergoing mitosis. Both LAD and non-LAD probes are present at similar distances from the DNA surface in cells in metaphase, a point in mitosis at which the nuclear lamina has disassembled (*Figure 6b*, *Figure 6—figure supplement 2*, *Video 2*, *Figure 6—source data 1*). However, by telophase, LAD probes have repositioned to the nuclear periphery (*Figure 6c*, *Video 3*, *Figure 6—source data 1*), indicating that H3K9me2-marked domains that were at the periphery in parent cells are specifically repositioned at the periphery in daughter nuclei before mitotic exit. In these same cells in telophase, non-LAD probes remained largely in the nucleoplasm, away from the nuclear lamina (*Figure 6c*, *Video 3*). Thus, specific LADs found at the nuclear periphery in parental cells are repositioned at the periphery at mitotic exit.

## Discussion

Our results provide experimental support of a model for nuclear peripheral localization and mitotic inheritance of lamina-associated heterochromatin (*Figure 7*). We show that H3K9me2 marks chromatin domains that are specifically positioned at the nuclear lamina during interphase. In mitosis, these domains retain and are bookmarked by H3K9me2. H3S10 phosphorylation promotes release from the nuclear periphery, likely by masking the Lys9 dimethyl modification from recognition by its reader/tether (*Fischle et al., 2003*; *Wang and Higgins, 2013*; *Eberlin et al., 2008*). In late stages of mitosis, dephosphorylation of H3S10 unmasks bookmarked LADs which are then reassembled

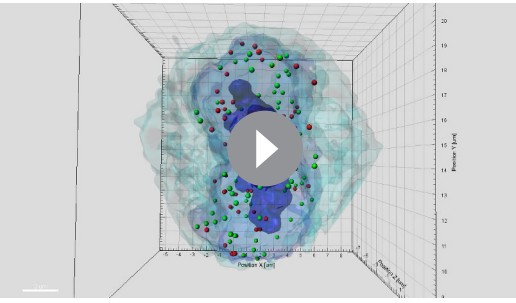

**Video 2.** 3D reconstruction of mESC in metaphase. Immunostained for Lamin B1 (cyan) and hybridized with fluorescent oligopaint probes for LADs (red) and non-LADs (green), and counterstained with DAPI (blue); pericentromeric heterochromatin displayed in dark blue.
DOI: https://doi.org/10.7554/eLife.49278.018

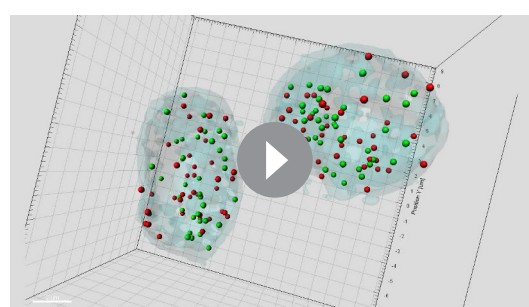

**Video 3.** 3D reconstruction of mESC in telophase. Immunostained for Lamin B1 (cyan) and hybridized with fluorescent oligopaint probes for LADs (red) and non-LADs (green), and counterstained with DAPI (blue).
DOI: https://doi.org/10.7554/eLife.49278.019

at the nuclear periphery during nuclear lamina reformation in the nuclei of daughter cells.

How cells convey information related to cellular identity to daughter cells has been a long-standing focus of investigation. Although mitotic chromosomes are condensed and transcriptionally silent, it is now appreciated that many nuclear factors remain associated with specific regions of mitotic chromatin, and some histone post-translational modifications are also retained. The concept of 'mitotic bookmarking' has been put forth to describe mechanisms by which transcriptionally active regions of euchromatin may be 'remembered' and rapidly re-activated upon mitotic exit (*Kadauke and Blobel, 2013*; *Palozola et al., 2019*; *Sureka et al., 2018*). Here, we extend this concept by elucidating a mechanism for transmitting a blueprint of the 3D organization of the genome from mother to daughter cell with a specific focus on peripheral heterochromatin associated with the inner nuclear lamina. Our data indicate that H3K9me2 acts as a 3D architectural mitotic guidepost.

Our results highlight the role of H3S10 phosphorylation adjacent to dimethylated Lys9 in 3D mitotic bookmarking. H3K9me2S10 phosphorylation allows for dissociation of peripheral heterochromatin from the nuclear lamina while retaining memory of genomic regions that will be reattached to the newly formed nuclear lamina upon dephosphorylation and mitotic exit. This example of a phospho-methyl switch extends previous studies that implicated related phospho-methyl switch mechanisms in transcriptional bookmarking without invoking regulation of 3D genome organization or nuclear reassembly. For example, H3S10 phosphorylation can displace HP1 binding to trimethylated Lys9 during mitosis (*Hirota et al., 2005*; *Fischle et al., 2005*). In another example, the active histone mark H3K4me3 is bound by TFIID and the basal transcriptional machinery during interphase. While H3K4me3 is maintained through mitosis, phosphorylation of Thr3 results in dissociation of TFIID and transcriptional silencing. The retention of H3K4me3 is thought to allow for rapid re-initiation of transcription after mitosis when Thr3 is dephosphorylated (*Varier et al., 2010*; *Sawicka and Seiser, 2014*). Our results supporting an H3K9me2S10 phospho-methyl switch suggest that this conserved mechanism also is employed for mitotic memory of nuclear architecture. During cell division, this mechanism is utilized to release all peripheral heterochromatin from the nuclear lamina, but it will be of interest to determine if a similar process occurs during interphase to release specific LADs from the periphery, perhaps endowing these domains with competence to be accessed by nuclear regulators of transcription. Histone phosphorylation, including H3S10, has been well documented to occur in response to classic signal transduction pathways such as Mapk signaling (*Winter et al., 2008*) suggesting a potential mechanism for the regulation of LAD release as a component of signal transduction.

The importance of the spatial organization of the genome has attracted increasing attention in recent years with a growing appreciation for unique, lineage-specific LADs and other architectural features. Largescale efforts have focused on characterizing genome organization in interphase, with less attention to how 3D architecture is transmitted through mitosis. Indeed, an early study suggested that LADs might be stochastically formed de novo following each cell division rather than inherited from the mother cell following mitosis (*Kind et al., 2013*). Unless all heterochromatic subcompartments are functionally equivalent, this would be somewhat inconsistent with the role that LADs are thought to play in cell identity (*Robson et al., 2016*; *Peric-Hupkes et al., 2010*; *Kohwi et al., 2013*; *Gonzalez-Sandoval et al., 2015*; *Poleshko et al., 2017*). Many reports have documented consistent, cell-type-specific LAD architecture as well as restoration of particular heterochromatin domains at the lamina after cell division (*Zullo et al., 2012*; *Kind et al., 2015*). It is conceivable that cell-type-specific LAD organization is 'rediscovered' after mitosis rather than 'remembered' and it has been reported that LADs can reshuffle between peripheral heterochromatin and perinucleolar heterochromatin. A recent study demonstrated that a subset of Nucleolus-Associated Domains (NADs) that exchange between nuclear lamina and nucleolar periphery are enriched for H3K9me3 (*Vertii et al., 2019*). Our results showing localization of H3K9me2-enriched lamina-associated chromatin, including those produced with LAD-specific oligopaints, suggest that H3K9me2-marked LADs which are re-established at the nuclear periphery at the end of mitosis concomitant with nuclear lamina re-assembly are likely distinct from the H3K9me3-marked NADs.

Mitosis and the period shortly following in G1 may provide a vulnerable period to regulate or modify genome organization. Consistent with this, pioneering experiments artificially tethering areas of the genome to the nuclear lamina noted the requirement for a mitotic event to precede efficient tethering of the genome to the nuclear lamina (*Finlan et al., 2008*; *Reddy et al., 2008*;

*Kumaran and Spector, 2008*). Moreover, nuclear transfer experiments demonstrated that mitotic chromatin can be reprogrammed to activate the core pluripotency network 100 times more efficiently than interphase chromatin (*Halley-Stott et al., 2014*). This may be, in part, because three-dimensional reorganization of the genome after mitosis helps to regulate accessibility. In particular, it is possible that the period during which H3S10 phosphorylation is lost in late mitosis, but before H3K9me2-marked chromatin is fully re-established as lamina-associated heterochromatin at the nuclear periphery, is a particularly vulnerable time to change LAD positioning in daughter cells. Hence, this may also coincide with a window in which cell fate changes associated with modifications in nuclear architecture occur (*Gilbert, 2010*). This would be in accord with the 'quantal theory of differentiation' put forth by Howard Holtzer over 50 years ago which proposed that major steps in lineage determination and cell fate restriction required mitotic events (*Holtzer et al., 1972*).

Classic cell biology experiments have demonstrated the necessity of kinase-phosphatase activity for mitotic progression and the requirement for chromatin to allow nuclear membranes to reform in daughter cells after mitosis (*Gerace and Blobel, 1980*; *Newport, 1987*; *Foisner and Gerace, 1993*; *Burke and Gerace, 1986*; *Wei et al., 1999*; *Prigent and Dimitrov, 2003*; *Wandke and Kutay, 2013*; *Haraguchi et al., 2008*). Our model provides a mechanistic explanation for these requirements and advances current models of mitotic bookmarking by introducing the concept of 3D architectural mitotic bookmarking. This model for epigenetic inheritance may have implications for understanding how cells adopt new fates in the setting of asymmetric cell divisions, and how cellular identity may be lost or altered in the context of cancer or trans-differentiation. For example, it will be of great interest to determine if the re-establishment of spatial chromatin organization is disrupted in cells as they undergo oncogenic transformation and/or cellular reprogramming.

# Materials and methods

## Key resources table

| Reagent type (species) or resource | Designation | Source or reference | Identifiers | Additional information |
|---|---|---|---|---|
| Strain, strain background (*C. elegans*) | WT | CGC | N2, RRID:WB-STRAIN:N2_(ancestral) | |
| Strain, strain background (*C. elegans*) | Cec-4 deletion | CGC | RB2301, RRID:WB-STRAIN:RB2301 | |
| Strain, strain background (*C. elegans*) | CEC4-mCherry transgene | *Gonzalez-Sandoval et al. (2015)* | GW849 | |
| Strain, strain background (*C. elegans*) | Cec-4 rescue with Cec-4-mCherry transgene | This paper | | |
| Cell line (*D. melanogaster*) | S2 | Maya Capelson lab | CVCL_TZ72, RRID:CVCL_TZ72 | Late embryonic stage cells |
| Cell line (*Xenopus laevis*) | S3 | Matthew Good lab | CVCL_GY00, RRID:CVCL_GY00 | Embryonic cells |
| Cell line (*Mus musculus*) | C2C12 | ATCC | CRL-1772, RRID:CVCL_0188 | C2C12 skeletal myoblast |
| Cell line (*Mus musculus*) | NIH/3T3 | ATCC | CRL-1658, RRID:CVCL_0594 | NIH/3T3 fibroblasts |
| Cell line (*Mus musculus*) | mESC | ATCC | CRL-1934, RRID:CVCL_4378 | Embryonic stem cells |
| Cell line (*Homo-sapiens*) | HeLa | ATCC | CCL-2, RRID:CVCL_0030 | |
| Cell line (*Homo-sapiens*) | IMR-90 | ATCC | CCL-186, RRID:CVCL_0347 | IMR-90 fibroblasts |

*Continued on next page*

*Continued*

| Reagent type (species) or resource | Designation | Source or reference | Identifiers | Additional information |
|---|---|---|---|---|
| Cell line (*Homo-sapiens*) | hESC | Rajan Jain lab | RRID:CVCL_EL23 | Induced pluripotent stem cells |
| Antibody | anti-H3K9me2 (Rabbit polyclonal) | Active Motif | Cat# 39239, RRID:AB_2793199 | IF (1:1000), WB (1:3000) |
| Antibody | anti-H3K9me2 (Rabbit polyclonal) | Active Motif | Cat# 39375, RRID:AB_2793234 | IF (1:1000) |
| Antibody | anti-H3K9me2 (Mouse monoclonal) | Abcam | Cat# ab1220, RRID:AB_449854 | IF (1:1000), WB (1:3000) |
| Antibody | Mouse anti-H3K9me2S10p | Active Motif | Cat# 61429, RRID:AB_2793632 | IF (1:1000) |
| Antibody | anti-H3K9me3 (Rabbit polyclonal) | Abcam | Cat# ab8898, RRID:AB_306848 | IF (1:1000) |
| Antibody | anti-H3K27me3 (Rabbit polyclonal) | EMD Millipore | Cat# 07–499, RRID:AB_310624 | IF (1:1000) |
| Antibody | anti-Lamin B1 (Rabbit polyclonal) | Abcam | Cat# ab16048, RRID:AB_10107828 | IF (1:1000) |
| Antibody | Goat anti-Lamin B (Goat polyclonal) | Santa Cruz | Cat# sc-6216, RRID:AB_648156 | IF (1:1000) |
| Antibody | Goat anti-Lamin B (Goat polyclonal) | Santa Cruz | Cat# sc-6217, RRID:AB_648158 | IF (1:1000) |
| Antibody | anti-Lamin A/C (Mouse monoclonal) | Santa Cruz | Cat# sc-376248, RRID:AB_10991536 | IF (1:1000) |
| Antibody | anti-LMN1 (Mouse monoclonal) | Developmental Studies Hybridoma Bank | Cat# LMN1, RRID:AB_10573809 | IF (1:1000) |
| Antibody | anti-histone H3 (Rabbit polyclonal) | Abcam | Cat# ab1791, RRID:AB_302613 | IF (1:1000) |
| Antibody | anti-GFP (Rabbit polyclonal) | Abcam | Cat# ab290, RRID:AB_303395 | IF (1:1000) |
| Antibody | anti-Rabbit AlexaFluor 555 (Donkey polyclonal) | Invitrogen | Cat# A31572, RRID:AB_162543 | IF (1:1000) |
| Antibody | anti-Rabbit AlexaFluor 488 (Donkey polyclonal) | Invitrogen | Cat# A21206, RRID:AB_2535792 | IF (1:1000) |
| Antibody | anti-Rabbit AlexaFluor 568 (Donkey polyclonal) | Invitrogen | Cat# A10042, RRID:AB_2534017 | IF (1:1000) |
| Antibody | anti-Rabbit AlexaFluor 647 (Donkey polyclonal) | Invitrogen | Cat# A31573, RRID:AB_2536183 | IF (1:1000) |
| Antibody | anti-Mouse AlexaFluor 488 (Donkey polyclonal) | Invitrogen | Cat# A21202, RRID:AB_141607 | IF (1:1000) |
| Antibody | anti-Mouse AlexaFluor 568 (Donkey polyclonal) | Invitrogen | Cat# A10037, RRID:AB_2534013 | IF (1:1000) |
| Antibody | anti-Goat AlexaFluor 488 (Donkey polyclonal) | Invitrogen | Cat# A11055, RRID:AB_2534102 | IF (1:1000) |
| Antibody | anti-Goat AlexaFluor 568 (Donkey polyclonal) | Invitrogen | Cat# A11057, RRID:AB_2534104 | IF (1:1000) |

*Continued on next page*

*Continued*

| Reagent type (species) or resource | Designation | Source or reference | Identifiers | Additional information |
|---|---|---|---|---|
| Antibody | anti-Goat AlexaFluor 647 (Donkey polyclonal) | Invitrogen | Cat# A21447, RRID:AB_2535864 | IF (1:1000) |
| Antibody | anti-Rabbit IgG, HRP-linked | Cell Signaling | Cat# 7074, RRID:AB_2099233 | WB (1:7500) |
| Antibody | anti-Mouse IgG, HRP-linked | Cell Signaling | Cat# 7076, RRID:AB_330924 | WB (1:7500) |
| Peptide array | MODified Histone Peptide Array | Active Motif | Cat# 13001 | |
| Peptide | H3K9me2 | Abcam | Cat# ab1772 | IF (1:500) |
| Peptide | H3K9me3 | Abcam | Cat# ab1773 | IF (1:500) |
| Peptide | H3K27me2 | Abcam | Cat# ab1781 | IF (1:500) |
| Peptide | H4K20me2 | Abcam | Cat# ab14964 | IF (1:500) |
| Peptide | H3K9me0 | EpiCypher | Cat# 12–0001 | IF (1:500) |
| Peptide | H3K9me1 | EpiCypher | Cat# 12–0010 | IF (1:500) |
| Peptide | H3K9me2 | EpiCypher | Cat# 12–0011 | IF (1:500) |
| Peptide | H3K9me3 | EpiCypher | Cat# 12–0012 | IF (1:500) |
| Peptide | H3K9me2S10p | EpiCypher | Cat# 12–0093 | IF (1:500) |
| Peptide | H3S10p | EpiCypher | Cat# 12–0041 | IF (1:500) |
| Recombinant DNA reagent | mEmerald-H3-23 (plasmid) | Addgene | Cat# 54115, RRID:Addgene_54115 | Histone H3 mEmerald-tag, deposited by Michael Davidson |
| Recombinant DNA reagent | H3 K9A (plasmid) | This paper | | Histone H3 with K9A substitution |
| Recombinant DNA reagent | H3 K9E (plasmid) | This paper | | Histone H3 with K9E substitution |
| Recombinant DNA reagent | H3 S10A (plasmid) | This paper | | Histone H3 with S10A substitution |
| Recombinant DNA reagent | H3 S10E (plasmid) | This paper | | Histone H3 with S10E substitution |
| Sequence-based reagent | H3 K9A forward | This paper | PCR primers | ACTAAACAGACAGCTCG GGCATCCACCGGCGGTAAAGCG |
| Sequence-based reagent | H3 K9A reverse | This paper | PCR primers | CGCTTTACCGCCGGTGGAT GCCCGAGCTGTCTGTTTAGT |
| Sequence-based reagent | H3 K9E forward | This paper | PCR primers | ACTAAACAGACAGCTCGGGA ATCCACCGGCGGTAAAGCG |
| Sequence-based reagent | H3 K9E reverse | This paper | PCR primers | CGCTTTACCGCCGGTGGATT CCCGAGCTGTCTGTTTAGT |
| Sequence-based reagent | H3 S10A forward | This paper | PCR primers | ACTAAACAGACAGCTCGGAAAG CCACCGGCGGTAAAGCG |
| Sequence-based reagent | H3 S10A reverse | This paper | PCR primers | CGCTTTACCGCCGGTGGCTTTC CGAGCTGTCTGTTTAGT |
| Sequence-based reagent | H3 S10E forward | This paper | PCR primers | ACTAAACAGACAGCTCGGAAAG AAACCGGCGGTAAAGCG |
| Sequence-based reagent | H3 S10E reverse | This paper | PCR primers | CGCTTTACCGCCGGTTTCTTTC CGAGCTGTCTGTTTAGT |
| Commercial assay or kit | QuikChange II XL Site-Directed Mutagenesis Kit | Agilent technologies | Cat# 200521 | |
| Software, algorithm | Imaris 9.0.1 | Bitplane | RRID:SCR_007370 | http://www.bitplane.com/imaris/imaris |

*Continued on next page*

*Continued*

| Reagent type (species) or resource | Designation | Source or reference | Identifiers | Additional information |
| --- | --- | --- | --- | --- |
| Software, algorithm | Image J | National Institute of Health | RRID:SCR_003070 | https://imagej.net/ |
| Software, algorithm | Vutara SRX | Bruker Corporation | | https://www.bruker.com /products/fluorescence- microscopes/vutara-super -resolution-microscopy/ overview/srx-software- vutara-super-resolution.html |
| Software, algorithm | GraphPad Prism 8 | GraphPad Software | RRID:SCR_002798 | http://www.graphpad.com/ |

## Cell lines

Mammalian cell lines were obtained from the American Type Culture Collection: murine NIH/3T3 fibroblast (ATCC, cat#CRL-1658), murine C2C12 skeletal myoblast (ATCC, cat#CRL-1772), murine embryonic stem cell (ATCC, cat# CRL-1934), human IMR-90 fibroblast (ATCC, cat#CCL-186) and HeLa cells (ATCC, cat#CCL-2). Xenopus S3 cells were obtained from the Matthew Good lab (University of Pennsylvania). *Drosophila* S2 cells were obtained from the Maya Capelson lab (University of Pennsylvania). All cell lines tested negative for mycoplasma contamination. NIH/3T3, C2C12, IMR-90 and HeLa cells were maintained at 37°C in DMEM supplemented with 10% FetalPlex serum complex (Gemini, cat#100–602), penicillin, and streptomycin. Mouse ESCs were maintained at 37°C on a feeder layer of mitotically inactivated MEFs in DMEM with 15% FBS (Fisher Scientific #SH3007003) and ESGRO LIF (EMD Millipore, cat#ESG1106). Human ES cells were maintained at 37°C in Stem-MACS iPS-Brew XF media (Miltenyi Biotec GmbH, cat#130-104-368), supplemented with penicillin, and streptomycin. Xenopus S3 cells were maintained at 25°C in 66% L-15 media (Gibco, cat#11415–064) with 10% fetal bovine serum (Atlanta Biologicals, cat#S11550), sodium pyruvate, penicillin, and streptomycin.

## Plasmids, mutagenesis and transfection

Expression plasmids for Histone H3-mEmerald was received from Addgene (cat#54115, deposited by Michael Davidson). This plasmid was used to create Histone H3 tail mutant constructs: H3 K9A, H3 K9E, H3 S10A and H3 S10E using QuikChange II XL Site-Directed Mutagenesis Kit (Agilent technologies, cat#200521) according to manufacturer's instruction. Plasmid transfections were performed with FuGENE 6 (Promega, cat#E2691) according to manufacturer instructions. For confocal imaging cells were plated on coverslips (EMS, cat#72204–01), then transfected at 50% confluency and fixed 48 hr post-transfection. Primers used for mutagenesis:

H3 K9A (5'-ACTAAACAGACAGCTCGGGCATCCACCGGCGGTAAAGCG, 5'-CGC TTTACCGCCGGTGGATGCCCGAGCTGTCTGTTTAGT); H3 K9E (5'-ACTAAACAGACAGC TCGGGAATCCACCGGCGGTAAAGCG, 5'-CGCTTTACCGCCGGTGGATTCCCGAGCTGTCTG TTTAGT); H3 S10A (5'-ACTAAACAGACAGCTCGGAAAGCCACCGGCGGTAAAGCG, 5'-CGC TTTACCGCCGGTGGCTTTCCGAGCTGTCTGTTTAGT); H3 S10E (5'-ACTAAACAGACAGC TCGGAAAGAAACCGGCGGTAAAGCG, 5'-CGCTTTACCGCCGGTTTCTTTCCGAGCTGTCTG TTTAGT).

## *C. elegans* strains, embryo cell isolation for immunofluorescence

The wild-type strain is N2; the *cec-4* null is deletion strain RB2301 from the Caenorhabditis Genetics Center (CGC); CEC4-mCherry transgene is the GW849 strain (gwSi17 [cec-4p::cec-4::WmCherry:: cec-4 3'UTR] II) obtained from Susan Gasser (*Gonzalez-Sandoval et al., 2015*). The rescue strain was created by crossing *cec-4* mutant [cec-4 (ok3124) deletion] males to GW849 hermaphrodites. Animals were grown as previously described (*Stiernagle, 2006*). For immunostaining, worms were bleached, then washed off the plate with M9 solution (86 mM NaCl, 42 mM $Na_2HPO_4$, 22 mM $KH_2PO_4$, and 1 mM $MgSO_4$, pH 6.5). They were washed with a bleach solution (15 ml MilliQ water, 4 ml Clorox, and 2 ml 5 M KOH) with shaking until adult bodies were dissolved. Then, embryos were washed twice with M9 solution, fixed with 4% formaldehyde solution (incubated at room

temperature (RT) for 15 min). Embryos were then flash frozen by immersing tube in an ethanol/dry ice bath for 2 min, thawed to RT, and then incubated on ice for 20 min and washed twice with PBS. Fixed embryos were spun on the coverslips at 1000 g for 10 min in cushion buffer (100 mM KCl, 1 mM $MgCl_2$, 0.1 mM $CaCl_2$, 10 mM HEPES pH7.7, 250 mM sucrose, 25% glycerol), then post-fixed with 2% PFA for 10 min at RT. A single-cell suspension of embryonic cells was prepared in a similar manner, but after the beach solution washing step embryos were washed three times in L15 media (Corning Cellgro, cat#10–045-CV), and then incubated in the 0.5 mg/ml Chitinase (Sigma, cat#C6137) in Boyd Buffer (25 mM HEPES pH 7.3, 118 mM NaCl, 48 mM KCl, 2 mM $CaCl_2$, 2 mM $MgCl_2$) at RT with rotation/aspiration to dissociate cells. Cells were pelleted at 1000 g for 5 min at 4°C and dissolved in PBS. Cells were kept at 4°C before immunostaining.

## Immunofluorescence

NIH/3T3 cells, C2C12 cells, IMR-90 cells, HeLa cells, undifferentiated mouse and human ES cells, *Xenopus laevis* S3 cells utilized for immunofluorescence experiments were grown on glass coverslips, fixed with 2% paraformaldehyde (PFA) (EMS, cat#15710) for 10 min at RT, washed 3 times with DPBS (Gibco, cat#14190–136), then permeabilized with 0.25% Triton X-100 (Thermo Scientific, cat#28314) for 10 min. After permeabilization, cells were washed 3 times with DPBS for 5 min, then blocked in 1% BSA (Sigma, cat#A4503) in PBST (DPBS with 0.05% Tween 20, pH 7.4 (Thermo Scientific, cat#28320)) for 30–60 min at RT. Incubated with primary antibodies for 1 hr at RT, then washed 3 times with PBST for 5 min. Incubated with secondary antibodies for 30–60 min at RT, then washed 2 times with PBST for 5 min. Samples were counterstained with DAPI solution (Sigma, cat#D9542) for 10 min at RT, then rinsed with PBS. Coverslips were mounted on slides using 80% glycerol mounting media: 80% glycerol (Invitrogen, cat#15514–011), 0.1% sodium azide (Sigma, cat#S2002), 0.5% propyl gallate (Sigma, cat#02370), 20 mM Tris-HCl, pH 8.0 (Invitrogen, cat#15568–025).

## Immunofluorescence and DNA oligo FISH

Mouse ESCs were grown on 0.1% porcine gelatin (Sigma, cat#G2500) coated glass coverslips (EMS, cat#3406), fixed with 2% PFA for 10 min at RT. Then cells were immunostained as described above. DNA oligo hybridization protocol was adopted from *Rosin et al. (2018)* (*Rosin et al., 2018*). In brief, after incubation with secondary antibodies, samples were washed with DPBS and post-fixed with 2% PFA for 10 min at RT, washed 3 times with DPBS and permeabilized with 0.7% Triton X-100 for 10 min at RT, then rinsed with DPBS. Incubate coverslips in 70% ethanol, 90% ethanol, and 100% ethanol for 2 min each, then incubate in 2X SSC (Corning, cat#46–020 CM) for 5 min. Incubate coverslips in 2X SSCT (2X SSC with 0.1% Tween) for 5 min at RT, then incubate in 2X SSCT with 50% Formamide for 5 min at RT. DNA denaturation was performed in 2X SSCT with 50% Formamide for 2.5 min at 92°C, then additional 20 min at 60°C. After DNA denaturation, samples were cooled to RT in humid conditions for 2–3 min, then hybridized with DNA oligo probes in ~50–100 pmol primary DNA probe. Coverslips were heated at 92°C for 2.5 min on a heat block. Samples were hybridized with DNA oligo probes overnight at 37°C in a humid chamber. After hybridization with primary DNA oligo probes samples were washed in 2X SSCT for 15 min at 60°C, then for 10 min in 2X SSCT for 10 min at RT, then transferred in 2X SSC for 5 min. Next samples were hybridized with a secondary fluorescent DNA oligo probes in dark humidified chamber for 3 hr at RT. Hybridization mix: 10% Formamide, 10% dextran sulfate, 10 pmol secondary DNA probe. After secondary hybridization samples were washed for 5 min in 2X SSCT at 60°C, then 2X SSCT at RT, and 2X SSC buffer with DAPI. Samples were rinsed with DPBS and mounted on a slide.

## Image acquisition

All confocal immunofluorescent images were taken using a Leica TCS SP8 3X STED confocal microscope using 63x/1.40 oil objective. DAPI staining (blue channel) were acquired using a PMT detector with offset −0.1%. All other staining (green, red and far red channels) were acquired using HyD detectors in the standard mode with 100% gain. All images were taken with minimal laser power to avoid saturation. 3D images were taken as Z-stacks with 0.05 μm intervals with a range of 80–250 Z-planes per nucleus. Confocal 3D images were deconvoluted using Huygens Professional software using the microscope parameters, standard PSF and automatic settings for background estimation. Stochastic Optical Reconstruction Microscopy (STORM) images were obtain using Vutara SRX

STORM system. Cells for STORM imaging were plated on confocal plates (MatTek, cat#P35GC-1.5–14 C). After immunostaining cells were kept in DPBS until image acquisition. STORM imaging was performed in fresh imaging buffer (50 mM Tris-HCl, pH 8.0, 10 mM NaCl, 10% (w/v) glucose (Sigma, cat#G8270), 1.5 mg MEA (Sigma, cat#30070), 170 AU Glucose oxidase (Sigma, cat#G2133), 1400 AU Catalase (Sigma, cat#C40)). Confocal channel shift alignment and STORM point spread function (PSF) calibration and channel shift alignment were performed using 0.1 µm TetraSpeck fluorescent beads (Invitrogen, cat#T7279).

## Image analysis

Image analysis were performed using Image J, Imaris 9.0.1, and Vutara SRX software. Representative confocal images show a single focal plane. 2D image analysis was performed using Image J software (National Institute of Health, USA). Line signal intensity profile plots were created using Plot Profile tool. Measurement of localization of the IF signal at the nuclear periphery was performed as a proportion of the signal at the nuclear periphery measured using a mask of the nuclear lamina or H3K9me2 signals to total signal in the nucleus. 3D image reconstructions were performed using Imaris 9.0.1 software (Bitplane AG, Switzerland) as described (*Poleshko et al., 2017*). In brief, nuclear lamina, nuclear DNA volume, and H3K9me2-marked chromatin structure were created using Surfaces tool with automatic settings based on the fluorescent signals from the anti-Lamin B, DAPI staining, and anti-H3K9me2 antibodies. DNA oligo FISH probe spots were generated using the Spots tool with a 250 nm diameter, created at the intensity mass center of the fluorescent probe signal. Distance from the center of the DNA oligo FISH spot to the edge of the nuclear lamina surface was quantified using the Distance Transformation tool. The thickness of the peripheral heterochromatin layer in mESC was calculated previously (*Poleshko et al., 2017*) as the distance from the H3K9me2 surface inner edge to nuclear lamina inner edge again using the Measurement Points tool. If the distance from the DNA oligo FISH spot to the nuclear lamina was smaller than (or equal to) the average thickness of peripheral chromatin, then the spot was counted as localized to nuclear periphery. In cases when the DNA oligo FISH signal was imbedded into the nuclear lamina layer, the measurement returned negative distances. STORM image and cluster analysis were performed using Vutara SRX software (Bruker, USA) and Voronoi Tessellation Analysis of H3K9me2 STORM images was performed in MATLAB 2016a in a fashion similar to *Andronov et al. (2016)* (*Andronov et al., 2016*). First, the lateral x,y localizations were input into the 'delaunayTriangulation' function, and then used to construct Voronoi polygons using the 'Voronoidiagram' function. Areas of the Voronoi polygons were determined from the vertices with the function 'polyarea'. Multiscale segmentation of the STORM images was carried out using an automatic thresholding scheme in which the thresholds were defined by comparing the Voronoi area distribution of the localizations to a reference distribution of the expected Voronoi areas of random coordinates drawn from a spatial uniform distribution (*Levet et al., 2015*). The reference distribution was estimated with a Monte-Carlo simulation. The first threshold was selected as $\rho=\delta$, where $\rho$ is the threshold and $\delta$ is the average Voronoi area for a uniform distribution of localizations. After applying this first threshold, the intersection between the Voronoi polygon area distribution and the distribution of Voronoi polygon areas corresponding to the Monte Carlo simulation was identified and applied as the second threshold. This procedure was iterated multiple times to define several thresholds at increasing density.

## Antibody validation

To test anti-H3K9me2 antibodies specificity for immunofluorescence assay, a set of short peptides mimicking histone tail lysine methylation was used. H3K9me2 antibodies were preincubated with blocking peptides according to manufacturer's recommendations (1 µg of the antibody with 1–2 µg of a peptide) in 1 ml of antibody blocking buffer (1% BSA in PSBT), then used for immunostaining. Anti-H3K9me2S10p antibody was tested on a MODified Histone Peptide Array (Active Motif, Cat#13001), anti-H3K9me2 antibodies were tested previously (*Poleshko et al., 2017*). Array analysis software (Active Motif) were use for analysis and graphical representation. Western blot using acid extracted histone (according to the manufacturer's protocol, Abcam) from C2C12 cells using anti-H3K9me2 antibodies demonstrated a single band corresponding to the histone H3.

## DNA oligo FISH probe design and generation

Target regions were based on constitutive LADs (LADs) or constitutive inter-LADs (non-LADs) as previously defined (*Meuleman et al., 2013*). For LADs, regions were selected only if they were also defined as LADs according to both LaminB and H3K9me2 ChIP-seq data from *Poleshko et al. (2017)*; for non-LADs, regions were selected only if they were also defined as non-LADs according to both LaminB and H3K9me2 ChIP-seq data from *Poleshko et al. (2017)*. Two to three of each, LAD and non-LAD, regions per mouse autosome were chosen for generation of DNA oligo libraries (*Supplementary file 1*). Oligopaint libraries were designed using the OligoMiner pipeline (*Beliveau et al., 2018*). Sequences of 42 nucleotides of homology to the regions of interest were mined from the mouse mm9 genome build using the default parameters of OligoMiner. Each probe was designed to target a 250 kb region of sequence at a density of 4 probes/kb when possible. Single stranded probes were produced using PCR, T7 RNA synthesis, and reverse transcription as described previously (*Rosin et al., 2018*).

## Western blot

Lysates were run on 4–12% Bis-Tris protein gels (Invitrogen #NP0335) and blots were probed with anti-H3K9me2 (Active Motif #39239, 1:3000 and Abcam #ab1220, 1:3000), anti-GFP (Abcam #ab290, 1:5000) or anti-H3 (Abcam #ab1791, 1:7500) primary antibodies according to the instructions of the manufacturer. Anti-rabbit or anti-mouse HRP-conjugated secondary antibodies (Cell Signaling #7074, #7076) were used at 1:7500. Visualization was achieved using ECLPrime (GE Life Sciences #RPN2232).

## ChIP-seq tracks

The accession number for the ChIP-seq data referenced (*Poleshko et al., 2017*) is NCBI GEO: GSE97878.

## Statistical analysis

Statistical analyses were performed with Graphpad PRISM 8.0.1 software (Graphpad Software, Inc) using ANOVA one-way non-parametric (Kruskal-Wallis) test with Dunn's multiple comparison or unpaired non-parametric Student's t-test (Mann-Whitney).

# Acknowledgements

We thank Andrea Stout from the Penn CDB Microscopy Core for help with imaging. We thank Matt Good, Nicolas Plachta and Gerd Blobel for discussions and comments on the manuscript. This work was supported by NIH (R35 HL140018 to JAE, DP2-HL147123 to RJ, and R35 GM127093 to JIM) and the Cotswold Foundation (to JAE), the WW Smith endowed chair (to JAE), Burroughs Welcome Career Award for Medical Scientists and the Gilead Research Scholars Program (to RJ). RJ and JAE received support from the NSF (CMMI-1548571).

# Additional information

### Funding

| Funder | Grant reference number | Author |
| --- | --- | --- |
| National Institutes of Health | R35 HL140018 | Jonathan A Epstein |
| National Institutes of Health | DP2-HL147123 | Rajan Jain |
| National Institutes of Health | R35 GM127093 | John Isaac Murray |
| Cotswold Foundation | | Jonathan A Epstein |
| WW Smith Endowed Chair | | Jonathan A Epstein |
| Burroughs Wellcome Fund | Career Award for Medical Scientists | Rajan Jain |
| National Science Foundation | CMMI-1548571 | Rajan Jain<br>Jonathan A Epstein |

| Gilead Sciences | Gilead Research Scholars Program | Rajan Jain |

The funders had no role in study design, data collection and interpretation, or the decision to submit the work for publication.

## Author contributions

Andrey Poleshko, Conceptualization, Data curation, Formal analysis, Validation, Investigation, Visualization, Writing—original draft, Writing—review and editing; Cheryl L Smith, Conceptualization, Investigation, Writing—original draft, Writing—review and editing; Son C Nguyen, Resources, Methodology; Priya Sivaramakrishnan, Resources; Karen G Wong, Investigation; John Isaac Murray, Resources, Funding acquisition; Melike Lakadamyali, Software, Methodology; Eric F Joyce, Resources, Funding acquisition, Methodology; Rajan Jain, Jonathan A Epstein, Conceptualization, Supervision, Funding acquisition, Writing—review and editing

## Author ORCIDs

Andrey Poleshko (iD) https://orcid.org/0000-0002-6656-2941
Cheryl L Smith (iD) https://orcid.org/0000-0003-4323-6382
Priya Sivaramakrishnan (iD) http://orcid.org/0000-0002-8717-8287
Karen G Wong (iD) https://orcid.org/0000-0003-2986-4091
John Isaac Murray (iD) http://orcid.org/0000-0003-4026-584X
Eric F Joyce (iD) http://orcid.org/0000-0002-0418-2804
Rajan Jain (iD) https://orcid.org/0000-0002-1979-044X
Jonathan A Epstein (iD) https://orcid.org/0000-0001-8637-4465

## Decision letter and Author response

Decision letter https://doi.org/10.7554/eLife.49278.028
Author response https://doi.org/10.7554/eLife.49278.029

# Additional files

## Supplementary files

• Supplementary file 1. Genomic coordinates (mm9) of regions targeted with oligopaint DNA probes.
DOI: https://doi.org/10.7554/eLife.49278.025

• Transparent reporting form
DOI: https://doi.org/10.7554/eLife.49278.026

## Data availability

All data generated or analysed during this study are included in the manuscript and supporting files.

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
