## [Decision Letter]

Thank you for submitting your article "H3K9me2 orchestrates inheritance of spatial positioning of peripheral heterochromatin through mitosis" for consideration by *eLife*. Your article has been reviewed by three peer reviewers, and the evaluation has been overseen by a Reviewing Editor and Jessica Tyler as the Senior Editor. The following individual involved in review of your submission has agreed to reveal their identity: Andrew S Belmont (Reviewer #3).

The reviewers have discussed the reviews with one another and the Reviewing Editor has drafted this decision to help you prepare a revised submission.

Summary:

This is a nice paper on an important topic that significantly advances mechanistic understanding of how nuclear lamina-associated domains of silenced chromatin can be remembered through mitosis and faithfully re-established in daughter cells. It also clarifies why/how H3K9me2 differs from H3K9me3 in its role and localization in cells. The authors address several fundamental points with regard to (a) the degree to which H3K9me2 marks peripheral heterochromatin; (b) the degree to which LADs are positioned specifically at the nuclear periphery versus also at other heterochromatin compartments in the nuclear interior, such as nucleoli or peri-centric heterochromatin; and (c) how LADs and these H3K9me2-marked domains position during the reformation of the nucleus after mitosis. The manuscript demonstrates that H3K9 di-methylation marks peripheral, lamina-associated heterochromatin through the cell cycle and proposes a phospho-methyl switch mechanism for displacing H3K9me2-lamina contacts during mitosis and resuming those contacts during nuclear reassembly. The authors propose that this allows the stable transmittance of 3D genome organization and faithful repression of LAD resident genes through cell divisions. This assertion would represent a significant and exciting advance to our understanding of how nuclear organization and thus cell-type-specific gene expression programs can be transmitted faithfully through cell division. The article is very well written and a pleasure to read. However, in its current status, there are questions that need to be resolved in order to proceed with acceptance of the manuscript.

Essential revisions:

1) The authors rightly describe the confusion in the field due to conflicting results in the literature concerning the distribution of the H3K9 mark relative to the nuclear periphery. They rightly attribute this to issues with regard to antibody specificity. But then they based their entire manuscript on exactly one antibody against H3K9me2. Given the importance of these results, and the existing conflicting data from many laboratories, it is important for the authors to validate their results with more than one antibody. The blocking peptide experiments are very valuable. But they just show that the antibody binds to the relevant peptide in solution. They do not address whether an epitope could be on a non-histone protein or if it recognizes the histone peptide sequence and modification but is also dependent on other properties surrounding the H3 histone in its native context.

2) How similar is the reported H3K9me2 staining relative to the previously described "epichromatin" epitope displayed by another anti-histone antibody by Don and Ada Olins? A direct comparison would be valuable. The similarity in comparing the figures in this manuscript with published papers from Olins and Olins is striking! This epichromatin staining is clearly context and conformation dependent – although the antibody is specific for histones, it only stains in situ the chromatin at the periphery of mitotic plates and at the nuclear periphery in nuclei, but this specificity is *not* shown for purified nucleosomes or sonicated chromatin using ChIP.

3) The ability of H3S10 phosphorylation to block peripheral tethering of H3K9me2-modified chromatin is supported by the inability of a phosphomimetic mutant of H3S10 (GFP-tagged H3S10E, Figure 3) to enrich at the nuclear periphery and by the anti-correlation between H3K9me2/S10 phosphorylation and lamina assembly around H3K9me2 (Figure 4B, 4C). However, in some images (Figure 4A) it seems that H3K9me2 and H3K9me2S10p both seem quite peripherally enriched through all stages of mitosis. The "increased" separation from the lamina actually seems to be just the visualization of a lower level of staining within the body of the condensed prophase chromatids. Are the line scans in Figure 4C single examples from one image? This point would be more convincing if the extent of H3K9me2's peripheral enrichment through mitosis were quantified similarly to the quantification done in Figure 3C. This would be complicated by the lack of an intact lamina to use as a fiducial mark, but the centroid of the chromatin mass could be used instead. Indeed, how do the authors reconcile the apparent contradiction between the peripheral staining of mitotic plates throughout mitosis using the H3K9me2 antibody (Figure 4) with the loss of peripheral localization until telophase for the LAD FISH (Figure 6)? This again opens the possibility that they are looking at an "epichromatin" like staining pattern with their antibodies.

4) These experiments use an antibody that recognizes the Histone H3 tail when dually modified with H3K9 dimethylation and H3S10 phosphorylation. This antibody is blocked by an H3K9me2S10p peptide but not by an H3K9me2 peptide. Is it blocked by an S10phospho peptide? If any S10phospho cross-reactivity exists, this may contribute central nucleoplasmic signal that may be more prominent especially as the specific antigen is removed (H3K9me2S10p). A different way to answer this question might be, how does S10phospho distribution compare to H3K9me2S10p distribution in mitotic cells? This may be an important point needed to support the argument that H3K9me2S10p must be de-phosphorylated for peripheral enrichment to resume.

5) Similarly, the authors have demonstrated the specificity of the H3K9me2 antibody for H3K9me2 over other methylation states. However, the ability of the H3K9me2 antibody to detect the H3K9me2S10p dual modification is not conclusively proven. For example, can the H3K9me2 antibody also be blocked by an H3K9me2S10p peptide? If not, this would suggest that this antibody has a lower affinity for the dually modified H3 tail than the H3K9me2/unmodified Ser10 tail. This would then open up the alternative interpretation that "enrichment" of H3K9me2 at the nuclear periphery over H3K9me2S10p is due to a higher affinity of the H3K9me2 antibody for un-phosphorylated H3 tails.

6) The idea of a phosphorus-switch by which H3K9me2S10 phosphorylation leads to loss of lamina association during mitosis, being a major punchline of the manuscript, does not appear to be demonstrated by the current manuscript. The argument used by authors rests on the lack of GFP-H3 localization of certain deletion mutants and some line scans of mitotic cells which were not convincing of showing loss of H3K9me2S10 phosphorylation from the peripheral staining. Overall, the authors show that H3K9me2 modification of genomic loci correlates with lamina association, and that histone S10 phosphorylation is anti-correlated with lamina association. However, functional tests to link these elements together are lacking. The authors assert that S10 phosphorylation disrupts the interaction of H3K9me2 with its tether. This could be tested, for instance, with Cec-4; if Cec-4 interacts specifically with H3K9me2 and not with H3K9me2S10p, this would support this model.

7) To what degree does the reduced z-resolution, projection through the depth of focus, and intensity scaling play a role in their conclusions of an exclusively peripheral localization of H3K9me2. Thus, in Results subsection “H3K9me2 is an evolutionarily conserved mark of peripheral heterochromatin”: "H3K9me2 marks only peripheral heterochromatin, whereas H3K9me3 and H3K27me3 co-localize with heterochromatin in the nuclear interior, or at both the interior and the periphery." However, the ratio of peripheral rim staining seems not that different for H3K9me2 and K3K9me3. If it is ignored the very intense staining over chromocenters in mouse cells that have large PCH, the ratio of the peripheral rim and internal foci staining does not seem that different for H3K9me2 and H3K9me3 staining. Eyeballing Figure 1A I see ratios of peripheral to interior foci intensities ranging from ~80:25 to 80-10 for H3K9me2, versus ~40:10 for H3K9me3 – a factor only of about 2-fold difference. Indeed, peripheral rim staining of chromatin in individual optical sections represents actually a z-projection through the z-depth of focus. Because of the finite thickness of the heterochromatin rim, this leads to a significant enhanced intensity due to this projection effect – as would be seen even for DNA staining. This effect is especially true for confocal imaging but also true for STORM imaging – both have much worse resolution in z. This effect needs to be compensated for when comparing the "enrichment" of signal at the periphery versus interior. The comparison could be with the corresponding measurement done for DNA staining such as DAPI for the peripheral rim and interior condensed foci (other than chromocenters) or between the intensity of internally stained foci with grazing sections of nuclei. Nuclei from cells growing flat in a monolayer will tend to have flat nuclear surfaces, particularly basal. These grazing sections will not have this superposition, projection effect. Finally, what is the actual comparison of intensity between the internal foci seen with H3K9me2 STORM staining and foci at the periphery. The beautiful STORM images in Figure 1B appear to show internal foci (spots and short fiber-like segments) at relatively the same brightness as foci at the periphery.

8) How exactly do the authors explain the GFP-H3 mutant results, given the documented low level of expression of the GFP-H3 variants? Can the authors elaborate on their logic? Thus, the H3K9me2 antibody rim staining appears unperturbed by any of the H3 mutants, suggesting that LAD distribution overall is unperturbed. Also, only a small fraction of the nucleosomes should contain the exogenous H3. But then why should this matter? Specifically, why should a 500-1000 kb LAD change position because a small percentage of the nucleosomes have the mutant H3? If there were actually some type of cooperative effect actually causing displacement, then why is the H3K9me2 staining unperturbed? Conversely, how would a modified nucleosome be able to localize 100s of nm or microns away from the nuclear periphery while the surrounding nucleosomes with wtH3 are localized at the periphery. These LADs contain condensed chromatin and its compaction and the known size of the nucleosome and linker DNA would seem to preclude such spatial separation.

[Editors' note: further revisions were requested prior to acceptance, as described below.]

Thank you for resubmitting your work entitled "H3K9me2 orchestrates inheritance of spatial positioning of peripheral heterochromatin through mitosis" for further consideration at *eLife*. Your revised article has been favorably evaluated by Jessica Tyler as the Senior Editor, A. Aguilera as the Reviewing Editor, and three reviewers.

The manuscript demonstrates that H3K9 di-methylation marks peripheral, lamina-associated heterochromatin through the cell cycle and proposes a phospho-methyl switch mechanism for displacing H3K9me2-lamina contacts during mitosis and resuming those contacts during nuclear reassembly. The authors propose that this allows the stable transmittance of 3D genome organization and faithful repression of LAD resident genes through cell divisions. The study contributes an important mechanistic understanding that significantly advances the field of epigenetics. In the revised manuscript, the authors have addressed all major concerns by including additional data to support antibody specificity, additional images demonstrating positioning of H3K9me2 through mitosis, and by pointing out that it was already demonstrated that Cec-4:H3K9methyl binding is disrupted by Ser10 or Thr11 phosphorylation, a point that solidifies the phospho-methyl switch model.

The manuscript has certainly been improved but there are some remaining issues that need to be addressed before acceptance, as outlined below:

The way the manuscript is now written seems to be more focused on the phospho-switch, the exclusive localization of the H3K9me2 to the periphery, and the absolute requirement for the H3K9me2 for peripheral localization in mammalian cells, the latter a conclusion entirely based on the GFP-H3 mutant localization data. However, the explanation of how the GFP-H3 mutants can be interpreted if they are only a small fraction of the H3 in the cell is not clear and that the authors should address this point with a detailed explanation in the paper itself – taking account of the low percentage of H3 that is GFP-tagged and mutated. The authors acknowledge that either biased incorporation or biased positioning of nucleosomes at/away from the nuclear periphery could explain their results. Anyhow, biased incorporation would represent a very different process than biased positioning, one that is completely different from the way that endogenous assembled nucleosomes are regulated by histone modifications. The authors should provide an explanation of how a low fraction of mutant H3 incorporation could mislocalize LADs from the periphery and explain the absence of any change to the endogenous H3K9me2 enrichment at the nuclear periphery.

With respect to the nonrandom binding of LADs after mitosis, the authors describe resolving a conflict in the literature regarding random shuffling of LADs to different heterochromatin compartments (i.e. nuclear lamina versus nucleoli) versus specific targeting of the same set of LADs to the periphery or interior (i.e. nucleolus) from mother to daughter cells, but it unclear they can do this by their current methodologies. It looks like that all can be concluded is that the preferred localization of ~80% of cLADs to the nuclear periphery occurs early after reformation of the telophase/early G1 nucleus. Please revise and discuss in more detail.

Figure 4. Typical confocal microscope hardware/software often has the user define a "black" level which is the analog level which the analog to digital (A/D) converter sets as the 0 value. All analog values below this "black" level are truncated to zero. If the "black" level is set to a level that represents non-zero intensity then this introduces a nonlinearity which prevents measurement of relative intensity levels. The hallmark of this is spatial resolution higher than possible with the psf of the microscope due to this truncation effect (i.e. values going from high to zero in a short distance relative to the normal blurring predicted by the point spread function) and also zero values of intensities inside the stained region and/or immediately outside of it. There is no description of how the authors set the "black" level during their microscopy. The images and line-scans indicate a black and zero level of intensity immediately outside the lamin ring staining and even at locations inside the nucleus. This is weird, as nonspecific antibody staining, out of focus light, even the in-focus point-spread function, and the dark current and readout noise typically produces nonzero intensity values. Therefore, the authors should describe how the intensity levels were set and whether they allow actual linear measurements of intensities.

Related to above, the predicted banding pattern of LADs versus iLADs could be better appreciated if they did chromosome spreads or looked at isolated chromosomes. This would tell if the unusual telomeric concentration of intensities towards the telomeres was real or not. If the staining of isolated chromosomes is different from the staining of the cells, it would point to a staining issue when staining whole cells.

Figure 4D. There is peripheral H3K9me2S10p staining although not a brighter ring of staining. It would be nice to see some type of aggregate analysis of a number of nuclei at each of several different stages of telophase to establish this temporal correlation.

---

## [Author Response]

Essential revisions:1) The authors rightly describe the confusion in the field due to conflicting results in the literature concerning the distribution of the H3K9 mark relative to the nuclear periphery. They rightly attribute this to issues with regard to antibody specificity. But then they based their entire manuscript on exactly one antibody against H3K9me2. Given the importance of these results, and the existing conflicting data from many laboratories, it is important for the authors to validate their results with more than one antibody. The blocking peptide experiments are very valuable. But they just show that the antibody binds to the relevant peptide in solution. They do not address whether an epitope could be on a non-histone protein or if it recognizes the histone peptide sequence and modification but is also dependent on other properties surrounding the H3 histone in its native context.

Localization of H3K9me2-marked chromatin at the nuclear periphery was previously confirmed with several independent antibodies and methods in our earlier publication (Poleshko et al., 2017) as mentioned in the text. In brief, localization of the H3K9me2-marked chromatin at the nuclear periphery was confirmed by immuno-fluorescence (IF) and ChIP-seq (see Figure 1—figure supplement 1B) using multiple validated antibodies. Furthermore, the OligoPaint experiment using probes to H3K9me2-enriched genomic regions correlates with the H3K9me2 IF staining (see Figure 4—figure supplement 1 and Figure 6—figure supplement 2).

To further address the reviewer’s concerns about antibody specificity, we have added a new Figure 2—figure supplement 1 that includes histone peptide array data for two H3K9me2 antibodies (Figure 2—figure supplement 1A-B), IF staining with 3 anti-H3K9me2 antibodies (Figure 2—figure supplement 1C) and a Western blot (Figure 2—figure supplement 1D). Note that IF staining with all 3 antibodies confirms the peripheral localization of H3K9me2-marked chromatin. Also, the Western blot shows only a single band at the appropriate size for H3, making recognition of a non-histone protein unlikely. The histone peptide arrays show that the antibodies recognize H3K9me2 in combination with other neighboring histone modifications, except S10p or T11p. This observation is consistent with the “phospho-methyl switch” model presented in the manuscript.

The histone peptide array data for the H3K9me2 antibodies were published previously and were referenced in the original text although some of the graphical representations shown in the revised figures are new. Given the reviewers’ concerns about antibody specificity, we believe that the manuscript will benefit from the added supplementary figure depicting this previously published data, which we have referenced appropriately in the figure legend.

2) How similar is the reported H3K9me2 staining relative to the previously described "epichromatin" epitope displayed by another anti-histone antibody by Don and Ada Olins? A direct comparison would be valuable. The similarity in comparing the figures in this manuscript with published papers from Olins and Olins is striking! This epichromatin staining is clearly context and conformation dependent – although the antibody is specific for histones, it only stains in situ the chromatin at the periphery of mitotic plates and at the nuclear periphery in nuclei, but this specificity is not shown for purified nucleosomes or sonicated chromatin using ChIP.

For current and previous studies we used two anti-H3K9me2 antibodies: Active Motif pAb #39239 and Abcam mAb #ab1220. Both antibodies were tested for specificity and the influence of neighboring modifications using a peptide array, and both antibodies showed a single, specific band by Western blot (new Figure 2—figure supplement 1). Peripheral localization of the H3K9me2-marked chromatin was confirmed by IF with 3 different antibodies. Given the extensive characterization of these antibodies and the confirmation of the peripheral location of H3K9me2-marked chromatin by multiple methods, we do not believe it would advance our work substantively to further explore the relationship of our work to the anti-nucleosome PL2-6 antibody used by Don and Ada Olins. Indeed, the cryoEM structure of the single chain antibody fragment from this antibody bound to a CENP-A nucleosome was recently published (Zhou et al., Nature Communications, 2019). Perhaps future studies from our lab, the Olins’ or others could address whether any connection exists between our findings and the pattern of staining seen with PL2-6 that recognizes a distinct epitope.

3) The ability of H3S10 phosphorylation to block peripheral tethering of H3K9me2-modified chromatin is supported by the inability of a phosphomimetic mutant of H3S10 (GFP-tagged H3S10E, Figure 3) to enrich at the nuclear periphery and by the anti-correlation between H3K9me2/S10 phosphorylation and lamina assembly around H3K9me2 (Figure 4B, 4C). However, in some images (Figure 4A) it seems that H3K9me2 and H3K9me2S10p both seem quite peripherally enriched through all stages of mitosis. The "increased" separation from the lamina actually seems to be just the visualization of a lower level of staining within the body of the condensed prophase chromatids. Are the line scans in Figure 4C single examples from one image? This point would be more convincing if the extent of H3K9me2's peripheral enrichment through mitosis were quantified similarly to the quantification done in Figure 3C. This would be complicated by the lack of an intact lamina to use as a fiducial mark, but the centroid of the chromatin mass could be used instead. Indeed, how do the authors reconcile the apparent contradiction between the peripheral staining of mitotic plates throughout mitosis using the H3K9me2 antibody (Figure 4) with the loss of peripheral localization until telophase for the LAD FISH (Figure 6)? This again opens the possibility that they are looking at an "epichromatin" like staining pattern with their antibodies.

We did not intend to suggest that H3K9me2 remains peripheral during all stages of mitosis and indeed, we did not state this. H3K9me2-/H3K9me2S10p-marked chromatin detaches from the nuclear lamina in prophase and is packed into mitotic chromosomes until the telophase stage when H3K9me2-chromatin is restored at the nuclear periphery (Figure 4). H3K9me2-chromatin and euchromatin remains distributed throughout the chromosome arms but is excluded from the center of the mitotic plate during prometaphase-metaphase. The center of the mitotic plate contains centromeres and pericentromeric heterochromatin enriched for H3K9me3 (Figure 5). We agree with the reviewer that a single confocal plane might not be definitive. Therefore, we have included a new Figure 4—figure supplement 1 that shows 3D image reconstructions of the images presented in Figure 4A.

Confocal images displayed in Figure 4C are representative and line profiles show an analysis of a single line. To address the reviewer’s point, we have included additional images and line profiles as new Figure 4—figure supplement 3.

As mentioned above, H3K9me2 staining is distributed throughout the chromosome arms during prometaphase-anaphase (new Figure 4—figure supplement 1). OligoFISH data using LAD and non-LAD probes are consistent with this observation. We have included a new Figure 6—figure supplement 2 that displays the 3-dimensional distribution of H3K9me2-enriched LAD and non-LAD probes throughout the chromosome arms with exclusion from the center of the mitotic plate where pericentromeric heterochromatin/chromocenters localize. This correlates well with the distribution of the H3K9me2 mark as shown in Figure 4A and Figure 4—figure supplement 1. We also modified Video 2 to highlight the central position of chromocenters and non-central distribution of LAD and non-LAD probes.

4) These experiments use an antibody that recognizes the Histone H3 tail when dually modified with H3K9 dimethylation and H3S10 phosphorylation. This antibody is blocked by an H3K9me2S10p peptide but not by an H3K9me2 peptide. Is it blocked by an S10phospho peptide? If any S10phospho cross-reactivity exists, this may contribute central nucleoplasmic signal that may be more prominent especially as the specific antigen is removed (H3K9me2S10p). A different way to answer this question might be, how does S10phospho distribution compare to H3K9me2S10p distribution in mitotic cells? This may be an important point needed to support the argument that H3K9me2S10p must be de-phosphorylated for peripheral enrichment to resume.

The H3K9me2S10p antibody does not recognize H3S10p. We extended the peptide blocking experiment with addition of the H3S10p peptide as the reviewer suggested (Figure 4—figure supplement 2A). We also added histone peptide array data demonstrating antibody specificity of the H3K9me2S10p antibody (Figure 4—figure supplement 2B).

5) Similarly, the authors have demonstrated the specificity of the H3K9me2 antibody for H3K9me2 over other methylation states. However, the ability of the H3K9me2 antibody to detect the H3K9me2S10p dual modification is not conclusively proven. For example, can the H3K9me2 antibody also be blocked by an H3K9me2S10p peptide? If not, this would suggest that this antibody has a lower affinity for the dually modified H3 tail than the H3K9me2/unmodified Ser10 tail. This would then open up the alternative interpretation that "enrichment" of H3K9me2 at the nuclear periphery over H3K9me2S10p is due to a higher affinity of the H3K9me2 antibody for un-phosphorylated H3 tails.

The H3K9me2 antibody does not recognize the H3K9me2S10p epitope. We have added histone peptide array data (Figure 2—figure supplement 1B) demonstrating antibody specificity. The antibody cannot recognize H3K9me2 if neighboring S10 or T11 is phosphorylated. As the reviewer suggested, we also tested anti-H3K9me2 antibodies in IF assays to determine that binding is not blocked by H3K9me2S10p peptides (Figure 2—figure supplement 1E-F). Each antibody used is specific for its epitope.

We believe that any confusion is the result of partial colocalization of H3K9me2 and H3K9me2S10p staining in Figure 4. We interpret this finding to suggest that not every S10 adjacent to K9me2 is phosphorylated during mitosis. Note that during telophase the separation of the two epitopes becomes more clear (Figure 4D). To avoid confusion, we have modified the text to address this point (subsection “H3K9me2 persists through mitosis and associates with reassembling nuclear lamina in daughter cells at mitotic exit”):

“Our data suggest that not every histone H3 Ser10 adjacent to H3K9me2 is phosphorylated since we observe some overlap of staining with the H3K9me2 and H3K9me2S10p antibodies.”

6) The idea of a phosphorus-switch by which H3K9me2S10 phosphorylation leads to loss of lamina association during mitosis, being a major punchline of the manuscript, does not appear to be demonstrated by the current manuscript. The argument used by authors rests on the lack of GFP-H3 localization of certain deletion mutants and some line scans of mitotic cells which were not convincing of showing loss of H3K9me2S10 phosphorylation from the peripheral staining. Overall, the authors show that H3K9me2 modification of genomic loci correlates with lamina association, and that histone S10 phosphorylation is anti-correlated with lamina association. However, functional tests to link these elements together are lacking. The authors assert that S10 phosphorylation disrupts the interaction of H3K9me2 with its tether. This could be tested, for instance, with Cec-4; if Cec-4 interacts specifically with H3K9me2 and not with H3K9me2S10p, this would support this model.

In order to address this point, we have provided additional antibody validation data which demonstrates that phosphorylation of S10 adjacent to H3K9me2 (H3K9me2S10p) blocks recognition of the epitope by the H3K9me2 antibodies (Figure 2—figure supplement 1B, E-F).

The functional experiment suggested by the reviewer has been published previously (Gonzalez-Sandoval et al., 2015) and we have referenced this result in the text. Briefly, the Gasser lab demonstrated that S10 or T11 phosphorylation reduces K9 methylation-dependent CEC-4 binding by 75 or 105 times, respectively. Combined, these results support the “phospho-methyl switch” model.

“Indeed, experimental results from the Gasser lab demonstrated that CEC-4 binds methylated H3K9 peptides and this binding is reduced by 2 orders of magnitude if the adjacent Ser10 is phosphorylated (Gonzalez-Sandoval et al., 2015).”

7) To what degree does the reduced z-resolution, projection through the depth of focus, and intensity scaling play a role in their conclusions of an exclusively peripheral localization of H3K9me2. Thus, in Results subsection “H3K9me2 is an evolutionarily conserved mark of peripheral heterochromatin”: "H3K9me2 marks only peripheral heterochromatin, whereas H3K9me3 and H3K27me3 co-localize with heterochromatin in the nuclear interior, or at both the interior and the periphery." However, the ratio of peripheral rim staining seems not that different for H3K9me2 and K3K9me3. If it is ignored the very intense staining over chromocenters in mouse cells that have large PCH, the ratio of the peripheral rim and internal foci staining does not seem that different for H3K9me2 and H3K9me3 staining. Eyeballing Figure 1A I see ratios of peripheral to interior foci intensities ranging from ~80:25 to 80-10 for H3K9me2, versus ~40:10 for H3K9me3 – a factor only of about 2-fold difference. Indeed, peripheral rim staining of chromatin in individual optical sections represents actually a z-projection through the z-depth of focus. Because of the finite thickness of the heterochromatin rim, this leads to a significant enhanced intensity due to this projection effect – as would be seen even for DNA staining. This effect is especially true for confocal imaging but also true for STORM imaging – both have much worse resolution in z. This effect needs to be compensated for when comparing the "enrichment" of signal at the periphery versus interior. The comparison could be with the corresponding measurement done for DNA staining such as DAPI for the peripheral rim and interior condensed foci (other than chromocenters) or between the intensity of internally stained foci with grazing sections of nuclei. Nuclei from cells growing flat in a monolayer will tend to have flat nuclear surfaces, particularly basal. These grazing sections will not have this superposition, projection effect. Finally, what is the actual comparison of intensity between the internal foci seen with H3K9me2 STORM staining and foci at the periphery. The beautiful STORM images in Figure 1B appear to show internal foci (spots and short fiber-like segments) at relatively the same brightness as foci at the periphery.

We do not rely solely on the immunofluorescence experiments to conclude that H3K9me2-marked chromatin is localized at the nuclear periphery. Localization of H3K9me2-marked chromatin was observed at the nuclear periphery by multiple methods presented in the manuscript and previously published, including genome-wide ChIP-seq results demonstrating high LB1-H3K9me2 co-occupancy (Poleshko et al., 2017) as mentioned in the original text.

As stated in the text, H3K9me3 and H3K27me3 heterochromatin are observed in multiple regions of the nucleus in addition to the nuclear periphery thus both are non-specific to the nuclear periphery. In contrast, the H3K9me2 predominantly localized at the nuclear periphery. These observations are consistent between IF and ChIP-seq data (see Poleshko et al., 2017).

The described effect has no influence on image analysis/quantifications or any conclusions presented in the manuscript. To address the reviewer’s concern, we have provided additional supplementary images to display XY, XZ and YZ-projections as well as 3D image reconstruction of the H3K9me2 staining (Figure 1—figure supplement 1A).

To address the nature of the internal H3K9me2 foci, a blocking peptide was used to distinguish specific from background signal of the H3K9me2 antibody (Figure 2, Figure 1—figure supplement 2 and Figure 2—figure supplement 1). As mentioned in the original text, the signal in the nuclear interior is largely background as confirmed by both confocal and STORM microscopies.

8) How exactly do the authors explain the GFP-H3 mutant results, given the documented low level of expression of the GFP-H3 variants? Can the authors elaborate on their logic? Thus, the H3K9me2 antibody rim staining appears unperturbed by any of the H3 mutants, suggesting that LAD distribution overall is unperturbed. Also, only a small fraction of the nucleosomes should contain the exogenous H3. But then why should this matter? Specifically, why should a 500-1000 kb LAD change position because a small percentage of the nucleosomes have the mutant H3? If there were actually some type of cooperative effect actually causing displacement, then why is the H3K9me2 staining unperturbed? Conversely, how would a modified nucleosome be able to localize 100s of nm or microns away from the nuclear periphery while the surrounding nucleosomes with wtH3 are localized at the periphery. These LADs contain condensed chromatin and its compaction and the known size of the nucleosome and linker DNA would seem to preclude such spatial separation.

As the reviewer points out, the GFP-H3 mutants are expressed at relatively low levels, and they do not appear to alter endogenous H3K9me2 staining. We therefore do not think that they are displacing LADs, i.e. LADs do not change position. We interpret the inability of the K9A (and other) mutants to partition to the periphery to suggest that lysine 9 dimethylation is required for either incorporation into peripheral nucleosomes, or for retention within nucleosomes at the periphery. Perhaps interaction with a dimethyl reader at the periphery stabilizes and incorporates a K9 dimethylated histone H3 protein. We have added our interpretation of these results to further address this point.

“We interpret the inability of the K9A and K9E mutants to partition to the periphery to suggest that lysine 9 dimethylation is required for either incorporation into peripheral nucleosomes, or for retention within nucleosomes at the periphery.”

[Editors' note: further revisions were requested prior to acceptance, as described below.]

[…] The manuscript has certainly been improved but there are some remaining issues that need to be addressed before acceptance, as outlined below:The way the manuscript is now written seems to be more focused on the phospho-switch, the exclusive localization of the H3K9me2 to the periphery, and the absolute requirement for the H3K9me2 for peripheral localization in mammalian cells, the latter a conclusion entirely based on the GFP-H3 mutant localization data. However, the explanation of how the GFP-H3 mutants can be interpreted if they are only a small fraction of the H3 in the cell is not clear and that the authors should address this point with a detailed explanation in the paper itself – taking account of the low percentage of H3 that is GFP-tagged and mutated. The authors acknowledge that either biased incorporation or biased positioning of nucleosomes at/away from the nuclear periphery could explain their results. Anyhow, biased incorporation would represent a very different process than biased positioning, one that is completely different from the way that endogenous assembled nucleosomes are regulated by histone modifications. The authors should provide an explanation of how a low fraction of mutant H3 incorporation could mislocalize LADs from the periphery and explain the absence of any change to the endogenous H3K9me2 enrichment at the nuclear periphery.

We disagree that the focus of the manuscript has changed. In the revised manuscript, we haven’t changed any of our conclusions and we have not removed any data. We provided additional data as requested by the reviewers.

The conclusion that H3K9me2 acts as a 3D architectural mitotic guidepost is based on multiple experiments and observations and not solely on the GFP-H3 mutant localization.

Given that both wild-type GFP-tagged H3 and the S10A mutant GFP-tagged H3 proteins are incorporated and observed at the nuclear periphery, the most straight-forward conclusion is that only *certain* H3 mutants, namely those that preclude critical modifications, are not localized to the nuclear periphery. We clarify our reasoning in the text.

“Given that wild-type GFP-H3 is incorporated and observed at the nuclear periphery, we interpret the inability of the K9A and K9E mutants to partition to the periphery to suggest that lysine 9 dimethylation is required for either incorporation into peripheral nucleosomes, or for retention within nucleosomes at the periphery.”

Further, we do not observe any alteration of the endogenous H3K9me2 staining at the nuclear periphery upon expression of low levels of mutant H3 as we stated in the prior response, and we do not think that LADs are displaced from the nuclear lamina. The reviewer previously asked why we think LADs are being displaced in this experiment and we responded “We therefore do not think that they are displacing LADs, i.e. LADs do not change position”. We therefore do not understand why we are asked again why LADs are mislocalized. They are not.

With respect to the nonrandom binding of LADs after mitosis, the authors describe resolving a conflict in the literature regarding random shuffling of LADs to different heterochromatin compartments (i.e. nuclear lamina versus nucleoli) versus specific targeting of the same set of LADs to the periphery or interior (i.e. nucleolus) from mother to daughter cells, but it unclear they can do this by their current methodologies. It looks like that all can be concluded is that the preferred localization of ~80% of cLADs to the nuclear periphery occurs early after reformation of the telophase/early G1 nucleus. Please revise and discuss in more detail.

In order to avoid any confusion that we were “resolving a conflict”, and to indicate that our data support (but do not prove) the model in which H3K9me2-marked LADs are specifically repositioned at the nuclear periphery, we have substituted the word “suggest” in the Discussion:

“Our results showing localization of H3K9me2-enriched lamina-associated chromatin, including those produced with LAD-specific oligopaints, suggest that H3K9me2-marked LADs which are re-established at the nuclear periphery at the end of mitosis concomitant with nuclear lamina re-assembly are likely distinct from the H3K9me3-marked NADs.”

Figure 4. Typical confocal microscope hardware/software often has the user define a "black" level which is the analog level which the analog to digital (A/D) converter sets as the 0 value. All analog values below this "black" level are truncated to zero. If the "black" level is set to a level that represents non-zero intensity then this introduces a nonlinearity which prevents measurement of relative intensity levels. The hallmark of this is spatial resolution higher than possible with the psf of the microscope due to this truncation effect (i.e. values going from high to zero in a short distance relative to the normal blurring predicted by the point spread function) and also zero values of intensities inside the stained region and/or immediately outside of it. There is no description of how the authors set the "black" level during their microscopy. The images and line-scans indicate a black and zero level of intensity immediately outside the lamin ring staining and even at locations inside the nucleus. This is weird, as nonspecific antibody staining, out of focus light, even the in-focus point-spread function, and the dark current and readout noise typically produces nonzero intensity values. Therefore, the authors should describe how the intensity levels were set and whether they allow actual linear measurements of intensities.

To address the reviewer’s concern, we extended the Methods section that describes image acquisition (see below). Confocal images were taken using the HyD detectors. Only DAPI staining was acquired with a PMT detector with a “black” level defined as an offset -0.1%. Images were taken using minimal laser power to ensure there was no signal saturation. The concerns raised by the reviewer are not relevant to images taken with HyD detectors.

“All confocal immunofluorescent images were taken using a Leica TCS SP8 3X STED confocal microscope using 63x/1.40 oil objective. DAPI staining (blue channel) was acquired using a PMT detector with offset -0.1%. All other staining (green, red and far red channels) were acquired using HyD detectors in the standard mode with 100% gain. All images were taken with minimal laser power to avoid saturation. 3D images were taken as Z-stacks with 0.05μm intervals with a range of 80-250 Z-planes per nucleus. Confocal 3D images were deconvoluted using Huygens Professional software using the microscope parameters, standard PSF and automatic settings for background estimation.”

Related to above, the predicted banding pattern of LADs versus iLADs could be better appreciated if they did chromosome spreads or looked at isolated chromosomes. This would tell if the unusual telomeric concentration of intensities towards the telomeres was real or not. If the staining of isolated chromosomes is different from the staining of the cells, it would point to a staining issue when staining whole cells.

Based on the proposed mitotic spread experiments, we assume that the reviewer refers to prometaphase-anaphase staining. We do not rely on these images to draw any conclusions. Note that we show markedly different patterns of staining in telophase for H3K9me2 and several other histone marks (Figure 5) making artifactual staining exceedingly unlikely. We do not feel that staining of isolated chromosomes would add substantially to our work.

Figure 4D. There is peripheral H3K9me2S10p staining although not a brighter ring of staining. It would be nice to see some type of aggregate analysis of a number of nuclei at each of several different stages of telophase to establish this temporal correlation.

The proposed additional analyses will not change the overall conclusions and we do not feel that it would add any clarity to our manuscript.